# REASONING GYM: Reasoning Environments for Reinforcement Learning with Verifiable Rewards

**Zafir Stojanovski**[1][*] **Oliver Stanley**[1,2][*] **Joe Sharratt**[1][*] **Richard Jones**[1][*]

**Abdulhakeem Adefioye**[1] **Jean Kaddour**[3][†] **Andreas Köpf**[1][†]

[1]Open-Thought    [2]Scale AI    [3]University College London

 GitHub

## Abstract

We introduce REASONING GYM (RG), a library of reasoning environments for reinforcement learning with verifiable rewards. It provides over 100 data generators and verifiers spanning multiple domains including algebra, arithmetic, computation, cognition, geometry, graph theory, logic, and various common games. Its key innovation is the ability to generate virtually infinite training data with adjustable complexity, unlike most previous reasoning datasets, which are typically fixed. This procedural generation approach allows for continuous evaluation across varying difficulty levels. Our experimental results demonstrate the efficacy of RG in both evaluating and reinforcement learning of reasoning models.

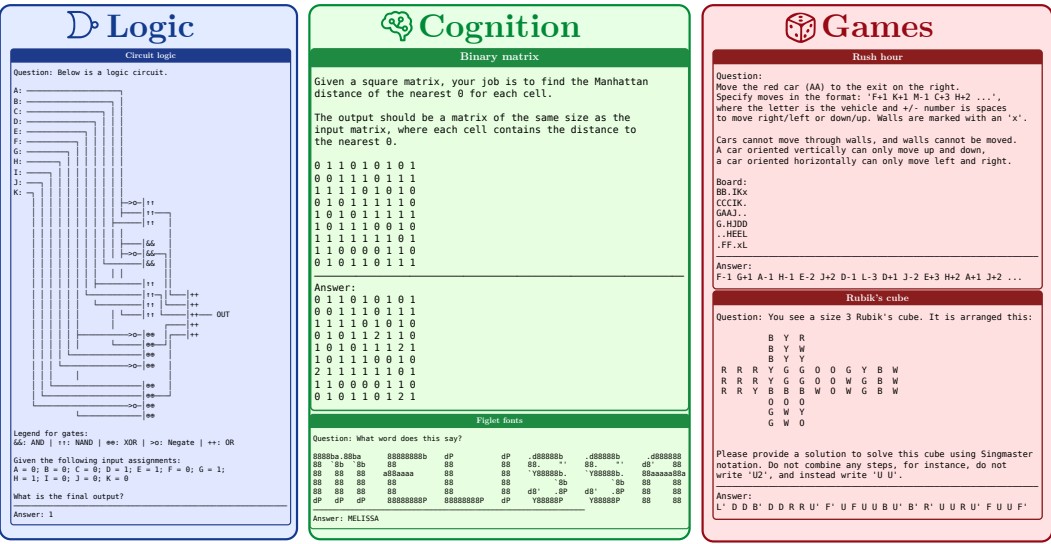

**Figure 1: Example  RG tasks from three categories.**

[*]Equal contribution, correspondence to zaf.stojano@gmail.com
[†]Equal advising

39th Conference on Neural Information Processing Systems (NeurIPS 2025) Track on Datasets and Benchmarks.

# 1  Introduction

The reasoning abilities of large language models (LLMs) have recently leapt forward, with models like OpenAI-o1 [49], DeepSeek-R1 [19], and QwQ-32B [72] setting new benchmarks. At the heart of this progress is Reinforcement Learning with Verifiable Rewards (RLVR) [36, 19], which leverages outcome-based feedback to unlock open-ended reasoning processes with diverse solution paths.

However, the success of RLVR hinges critically on the availability of high-quality training data. Current approaches face a fundamental scalability bottleneck [40]: they depend either on expensive human-curated question-answer pairs [11] or on internet-scraped content [46, 1] that is neither sustainable nor reliable in the long term [74, 29]. As reasoning models continue to advance, this data scarcity threatens to become an increasingly severe constraint on further progress.

We address this challenge with REASONING GYM (RG), a comprehensive library of procedurally generated [10] reasoning environments designed specifically for RLVR training. Unlike traditional reasoning benchmarks that provide fixed datasets, RG offers over 100 algorithmically verifiable tasks that can generate unlimited training instances with controllable difficulty and structural variation. These environments span diverse reasoning domains: symbolic algebra, discrete algorithms, spatial geometry, formal logic, pattern recognition, and constraint-based puzzles. Each task is equipped with verification mechanisms and parameters that enable fine-grained control over problem complexity.

The procedural nature of RG addresses several critical limitations of existing approaches. First, it eliminates memorization concerns by ensuring that no two generated instances are identical. Second, it enables dynamic curriculum learning, where task difficulty can be adjusted based on model performance. Third, it provides unlimited training data, removing the bottleneck imposed by fixed dataset sizes. Finally, it offers precise experimental control, allowing researchers to isolate specific reasoning capabilities and study their development systematically.

Our experimental investigation reveals several key insights:

- **Zero-shot performance of frontier LLMs is low for many RG tasks**, specifically the ones that represent visual concepts in text format like ARC, cognition, and games categories.
- **Increasing task difficulty creates sharp performance cliffs**. When transitioning from easy to hard configurations, performance drops are most severe in algorithmic reasoning (28%), code generation (62%), and graph problems (30%).
- **Larger non-reasoning models often underperform smaller reasoning models.** Performance drops are highest when transitioning from reasoning to non-reasoning models, underlining the value of reasoning data.
- **Curriculum RLVR results in improved final models**, with adaptive difficulty progression outperforming fixed-difficulty training across reasoning tasks.
- **RLVR generalizes across tasks from the same domain**, from mathematics to games. We observe this intra-domain improvement in both tasks the LLM is already competent in, as well as tasks the pre-RLVRed model fails to solve.
- **Signs of cross-domain transfer emerge from RLVR training.** A model trained on algorithmic tasks exhibits substantial improvements in math domains such as algebra and geometry.
- **Skills transfer to external benchmarks**. RLVR training on RG tasks improves performance on benchmarks such as MATH [21], GSM8K [11], Big-Bench Hard [70], and MMLU-Pro [79].

We release the complete library, including all task generators, training infrastructure, and experimental configurations, at https://github.com/open-thought/reasoning-gym/.

# 2  REASONING GYM (RG)

Despite rapid progress in language model reasoning, empirical work is bottlenecked by benchmarks that are either fixed in size, quickly memorized, or too noisy. Therefore, REASONING GYM (RG) is motivated by the need for an open-ended playground where models can be pushed past the "dataset ceiling", exposed to ever-harder instances, and evaluated with fully automatic, unambiguous rewards so that genuine reasoning improvements, and not dataset familiarity, drive the next wave of advances.

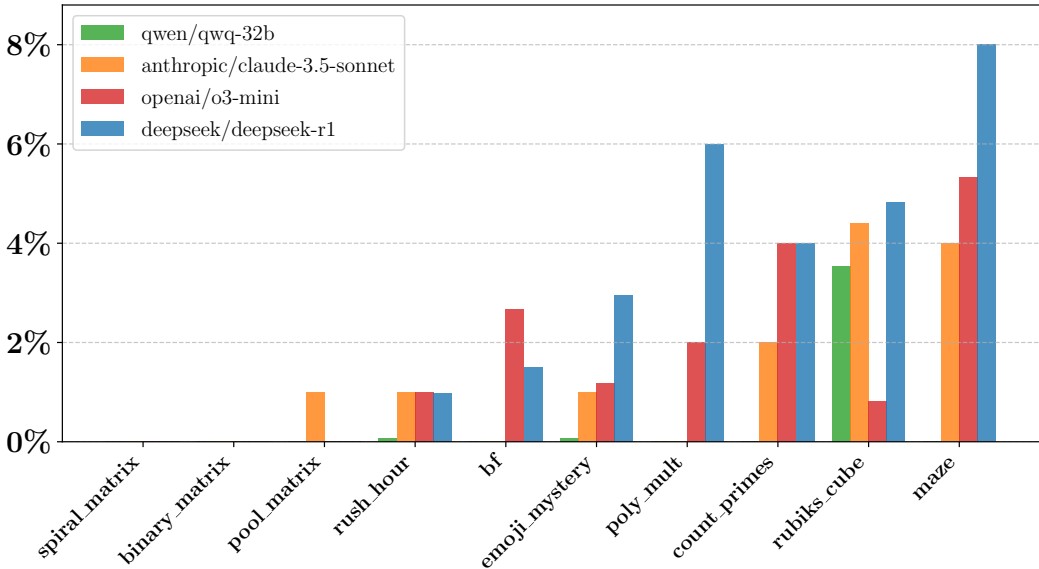

**Figure 2: Frontier models struggle with challenging 🌐 RG configurations.** Reasoning models like o3-mini [48] and DeepSeek-R1 [19] tend to outperform non-reasoning models, but the tasks configured with challenging parameters are still far from being saturated.

Following are the core design principles that underpin 🌐 RG:

(P1) **Algorithmic Verifiability.** Every task admits automatic verification and requires no human judgment. This enables reliable RLVR training while eliminating subjective evaluation.

(P2) **Large Solution Spaces.** Tasks are designed with expansive solution spaces, rewarding generalizable strategies above overfitting and mitigating reward hacking.

(P3) **Parametric Difficulty Control.** Configurable parameters systematically control problem characteristics, enabling dynamic curricula via precise difficulty adjustment.

To probe reasoning competence across a broad spectrum of skills, we partition 🌐 RG's generators into several high-level categories that mirror the abstractions humans rely on when solving problems:

- 🧮 **Mathematical domains**: algebra, arithmetic, geometry
- 🔀 **Algorithmic thinking**: search, optimization, procedures
- 🔖 **Logical reasoning**: formal proofs, inference rules
- 🧠 **Pattern recognition**: sequences, visual analogies
- 🎲 **Constraint satisfaction**: games, puzzles, planning

Figure 1 shows representative examples demonstrating the diversity of reasoning challenges, and Table 6 outlines the full set of categories alongside the data generators for each. We believe this taxonomy lets practitioners target specific abilities during training or evaluation while still drawing from a rich, procedurally generated mix of challenges.

Concretely, within each category we instantiate tasks not as fixed question-answer pairs, but as generative algorithms whose parameters continuously modulate problem characteristics:

- **Difficulty Parameters** directly control complexity (node counts for graphs, polynomial degrees for algebra, word lengths for language tasks).
- **Structural Parameters** determine fundamental problem properties (dimensionality, constraint types, proof depth).
- **Stylistic Parameters** vary presentation without affecting difficulty (variable names, number formats, problem framing).

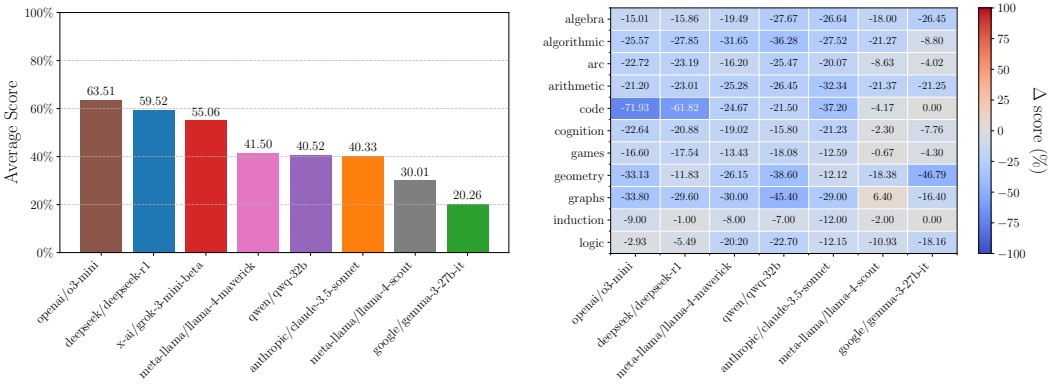

**(a)** Zero-shot performance of frontier LLMs.     **(b)** Difficulty Cliff: easy vs. hard tasks.

**Figure 3: Model and task difficulty comparison.** Left: Zero-shot ability across model types on the hard configs. Right: Impact of dataset difficulty on per-category accuracy. Section A.3 details the easy and hard parameter configurations for each dataset.

# 3 Zero-shot performance of LLMs

We conduct a comprehensive evaluation of state-of-the-art language models on 🌐 RG tasks, revealing challenges that persist even for frontier models. Our analysis encompasses both zero-shot capabilities and the effects of task difficulty scaling. Figures 7 and 8 report the precise scores for every task–model combination under the easy and hard settings, respectively.

## 3.1 Model Capabilities Across Reasoning Domains

Figures 2 and 3 present our core findings on model performance across 🌐 RG tasks. The results reveal a clear hierarchy among different model classes, with reasoning-optimized systems demonstrating substantial advantages over general-purpose alternatives.

**Reasoning vs. Non-reasoning Models.** The performance gap between reasoning-optimized and general-purpose models is striking and consistent. Models explicitly trained for reasoning, including o3-mini (63.5%), DeepSeek-R1 (59.5%), and Grok 3 Mini (55.1%), form a distinct leading group (Figure 3a). In contrast, strong general-purpose systems like Llama 4 Maverick (41.5%), Claude 3.5 Sonnet (40.3%), and Gemma 3 27B (20.3%) achieve substantially lower performance.

This 22% gap between the best reasoning and non-reasoning models represents more than a marginal improvement; suggesting that RLVR unlocks a step change in capabilities. The consistency of this advantage across 🌐 RG's diverse task categories indicates that reasoning-specific training develops broadly applicable skills rather than narrow-domain expertise.

**Performance Patterns Across Domains.** Examining performance by task category reveals interesting patterns in model capabilities. Mathematical domains (algebra, arithmetic, geometry) show relatively strong performance across all model types, likely reflecting the emphasis on mathematical reasoning in recent training regimes. However, tasks requiring visual-spatial reasoning represented in text format (cognition, games) prove particularly challenging, with even the strongest models achieving less than 50% accuracy.

Algorithmic tasks present an intermediate challenge, with clear performance differences between reasoning and non-reasoning models. This suggests that while basic algorithmic thinking is present in general-purpose models, the systematic problem decomposition required for complex algorithmic reasoning benefits significantly from specialized training.

## 3.2 The Difficulty Cliff Phenomenon

One of the most striking findings from our evaluation concerns the dramatic performance degradation when task difficulty increases. Figure 3b illustrates this phenomenon, showing how performance changes when transitioning from easy to hard task configurations.

Performance degradation is commonly observed uniformly across domains and model families. For o3-mini, the steepest declines occur in code ($-71.9\%$), graphs ($-33.8\%$), geometry ($-33.1\%$), and algorithms ($-25.6\%$). DeepSeek-R1 shows a similar pattern, with drops of $-61.8\%$, $-29.6\%$, $-11.8\%$, and $-27.9\%$ on the same categories, respectively. Overall, most model–task pairs exhibit notable performance declines as difficulty increases.

These results reveal implications that inform future research directions:

**Current models have shallow competencies.** The dramatic performance drops with increased difficulty suggest that current reasoning capabilities are more fragile than commonly assumed. Models may be learning to recognize and apply solution templates rather than developing robust reasoning strategies, which has also been indicated by concurrent work [85, 89, 78, 62, 41].

**Visual-spatial reasoning remains challenging.** Spatial reasoning in text-based representations proves particularly difficult for all models, as has also been shown by previous work [8, 9, 58].

**Domain-specific patterns exist.** The varying difficulty cliff magnitudes across domains indicate that reasoning challenges are not uniform. Some domains (like basic arithmetic) may be approaching saturation, while others (like complex algorithmic reasoning) remain largely unsolved.

# 4 Skill Transfer and Generalization

A central question in reasoning research concerns whether skills learned on specific tasks transfer to related problems. RG's diverse task categories provide an ideal testbed for investigating both intra-domain transfer (within reasoning categories) [13, 73, 22] and cross-domain transfer (across different types of reasoning) [37, 90].

In the experiments below, the training reward plots represent the *total reward*, computed as the sum of an *accuracy* component and an *auxiliary* component that rewards proper output formatting. By contrast, the evaluation tables report *only the accuracy component*, rescaled to a percentage in the range $0 - 100\%$. For transparency, we note that the reported results from the experiments require approximately 1500 A6000 hours, which we obtained by renting cloud GPUs from Runpod.

## 4.1 Intra-Domain Transfer

We first investigate whether RLVR training on a subset of tasks within a reasoning domain improves performance on held-out tasks from the same domain. This tests whether models develop domain-specific reasoning strategies that generalize beyond the specific tasks they were trained on.

**Experimental Design.** For each major reasoning category in RG, we trained Qwen2.5-3B-Instruct [84] using GRPO [63] on a composite of tasks from that category, then evaluated performance on a held-out task from the same domain. Each experiment involved three independent runs on identical evaluation sets of 50 problems, providing robust estimates of transfer effects.

**Training Dynamics.** Figure 4 illustrates the learning dynamics across different reasoning domains. Most categories exhibit rapid initial improvement, reflecting both format learning and genuine skill acquisition. The exception is arithmetic, where the base model already demonstrates strong competency, likely due to extensive mathematical training in its supervised fine-tuning phase. This ceiling effect provides a useful control, showing that our training improvements reflect genuine learning rather than artifacts.

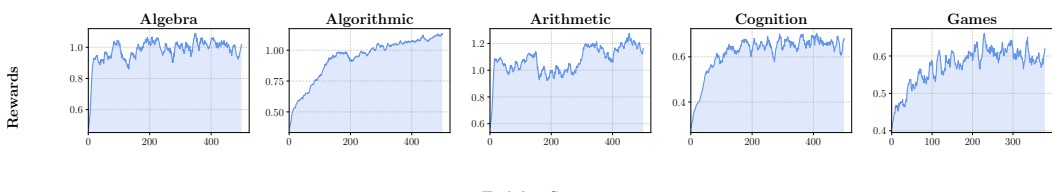

**Figure 4: Rewards of Intra-Domain Generalization RL**. There is a sharp increase in reward at the start of training. This is partly attributable to the model quickly learning auxiliary rewards (i.e. formatting) during training, but it is also reflective of how quickly RLVR improves the model's ability to solve training tasks.

**Table 1: Intra-Domain Generalization.** Acc@3 performance by dataset category. Baseline: Qwen2.5-3B-Instruct [84]; RG-RLVR: Qwen2.5-3B-Instruct [84] RL-fine-tuned on composite tasks from the particular category. Bold RG-RLVR scores are higher than Baseline. RLVR consistently improves performance across all tested categories covering diverse domains.

|  | Algebra | Algorithmic | Arithmetic | Cognition | Games |
|---|---|---|---|---|---|
| Baseline | 5.0 | 52.3 | 89.7 | 40.3 | 0.0 |
| RG-RLVR | **16.7**$^{+11.7}$ | **59.7**$^{+7.4}$ | **96.0**$^{+6.3}$ | **42.3**$^{+2.0}$ | **3.3**$^{+3.3}$ |

**Transfer Results.** Table 1 demonstrates consistent intra-domain transfer across all reasoning categories. The improvements range from modest gains in domains where the base model already shows competency (arithmetic: $+6.3\%$, cognition $+2.0\%$) to larger improvements in challenging domains (algebra: $+11.7\%$, algorithmic $+7.4\%$).

Particularly striking is the Games category, where the base model achieves zero accuracy but develops measurable capability ($3.3\%$) after RLVR training. This suggests that domain-specific training can bootstrap entirely new reasoning capabilities, not merely refine existing ones. The consistency of improvements across diverse difficulty levels indicates that RLVR develops robust domain-specific strategies rather than task-specific solutions.

## 4.2 Cross-Domain Transfer

More surprising than intra-domain transfer is the possibility that reasoning skills learned in one domain might benefit performance in entirely different domains. This would suggest that RLVR instills general reasoning capabilities that transcend specific problem types. Such transfer is critical because it indicates whether models develop fundamental reasoning primitives versus domain-specific heuristics. Demonstrating cross-domain transfer would validate that procedurally generated training data can develop broadly applicable reasoning skills rather than narrow task-specific competencies.

**Training Protocol.** We train separate instances of Qwen2.5-3B-Instruct [84] on individual 🌐 RG categories, then evaluate their performance on held-out tasks from different domains. This design isolates the effects of cross-domain transfer by ensuring that models never see data from the evaluation domains during training. Each cross-domain evaluation involves three independent runs to ensure robust estimates.

**Training Dynamics Across Domains.** Figure 5 reveals distinct learning patterns across reasoning domains. While most categories show sustained improvement throughout training, the Games category plateaus early, suggesting fundamental challenges in learning visual-spatial reasoning from text representations. This pattern provides insight into which reasoning skills are most amenable to current RLVR approaches.

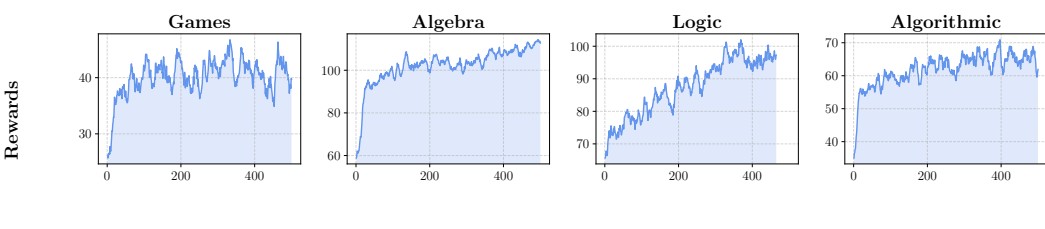

**Figure 5: Rewards of Cross-Domain Generalization RL.** Rewards initially spike due to learning the format reward (worth 0.2, with an accuracy reward worth 1.0). The model is then able to learn in all cases, but the differing trajectories and final reward values illustrate that some task categories are more challenging than others.

Table 2: **Cross-Domain Generalization.** Acc@3 performance by dataset category. Baseline: Qwen2.5-3B-Instruct [84]. RG-X: Qwen2.5-3B-Instruct [84] RL-fine-tuned on category X tasks. Bold RG-X scores are higher than Baseline. RLVR leads to notable performance improvements across domains.

| Test Dataset | Baseline | RG-Algebra | RG-Algorithmic | RG-Logic | RG-Games |
|---|---|---|---|---|---|
| Algebra | 23.83 | — | **52.89**+29.1 | — | **45.61**+21.8 |
| Algorithmic | 13.49 | **20.68**+7.2 | — | **16.43**+2.9 | — |
| ARC | 6.49 | 4.18-2.3 | — | — | 4.26-2.2 |
| Arithmetic | 29.56 | **46.14**+16.6 | **45.17**+15.6 | — | — |
| Cognition | 11.62 | — | — | **24.94**+13.3 | **24.71**+13.1 |
| Games | 8.40 | **9.23**+0.8 | — | 7.64-0.8 | — |
| Geometry | 0.83 | — | **23.17**+22.3 | — | **6.83**+6.0 |
| Graphs | 19.81 | — | — | **28.86**+9.1 | **22.49**+2.7 |

**Cross-Domain Transfer Results.** Table 2 reveals remarkable patterns of Cross-Domain transfer that exceed our initial expectations. Several key findings emerge:

- **Algorithmic training transfers broadly**: Models trained on algorithmic tasks show substantial improvements in algebra (+29.1%) and geometry (+22.3%), suggesting that procedural reasoning skills generalize across mathematical domains.
- **Logic training enhances pattern recognition**: Training on logic tasks improves performance in cognition (+13.3%) and graph reasoning (+9.1%), indicating shared underlying reasoning mechanisms.
- **Games training shows selective transfer**: Despite poor in-domain performance, games-trained models improve on algebra (+21.8%) and cognition (+13.1%), suggesting that constraint satisfaction skills transfer to other domains.

These results provide strong evidence that RLVR training develops transferable reasoning capabilities that extend far beyond the specific domains where training occurs.

## 4.3 Transfer to External Benchmarks

The ultimate test of 🌐 RG's utility lies in whether skills developed through training on procedurally generated tasks transfer to established reasoning benchmarks. We investigate this by training models on algorithmic and mathematical 🌐 RG categories, and then evaluating performance widely-used benchmarks for reasoning and understanding. In particular we use GSM8K [11], MATH [21] and Big-Bench Hard [70] for evaluating mathematical and logical reasoning; and MMLU-Pro [79] for advanced knowledge across academic and professional domains.

**Experimental Protocol.** The RG-Math model is Qwen2.5-3B-Instruct [84] trained for 800 GRPO [63] steps on a composite of algebra, arithmetic, and geometry tasks from 🌐 RG. The RG-Algorithmic model is the same checkpoint from Section 4.2 (Cross-Domain Transfer). All evaluations were performed with the Language Model Evaluation Harness [15] to ensure standardized comparison.[84]

Table 3: **External Generalization on GSM8K [11], MATH [21], and Big-Bench Hard [70].** Baseline: Qwen2.5-3B-Instruct [84]; RG-Math: Qwen2.5-3B-Instruct RL-fine-tuned on composite math tasks. Bold RG-X scores are higher than Baseline. RLVR on RG data leads to improvements across the challenging set of established benchmarks.

| Model | GSM8K [11] 8-shot, CoT | | MATH [21] 0-shot, CoT | | Big-Bench Hard [70] 3-shot, CoT | |
|---|---|---|---|---|---|---|
| | Score | Std Error | Score | Std Error | Score | Std Error |
| Baseline | 76.2 | 1.17 | 48.5 | 0.68 | 8.68 | 0.30 |
| RG-Math | **76.7**+0.5 | 1.16 | **58.2**+9.7 | 0.66 | **16.34**+7.66 | 0.40 |

**Table 4: External generalization on tasks from MMLU-Pro [79].** Baseline: Qwen2.5-3B-Instruct [84]; RG-X: Qwen2.5-3B-Instruct [84] RL-fine-tuned on category X tasks. Bold RG-X scores are higher than Baseline. Both RLVR on RG data lead to general improvements, with higher gains observed from the RG-Math model.

| Task | Baseline | | RG-Algorithmic | | RG-Math | |
|---|---|---|---|---|---|---|
| | Score | Std Error | Score | Std Error | Score | Std Error |
| Math | 54.63 | 1.35 | 53.89 -0.74 | 1.36 | **60.25** +5.62 | 1.33 |
| Computer Science | 37.80 | 2.40 | 40.73 +2.93 | 2.43 | **42.20** +4.40 | 1.47 |
| Physics | 38.49 | 1.35 | 39.26 +0.77 | 1.36 | **44.19** +5.70 | 1.38 |
| Engineering | 28.28 | 1.45 | **31.48** +3.20 | 1.49 | **31.17** +2.89 | 1.49 |
| Economics | 50.59 | 1.72 | 53.44 +2.85 | 1.72 | **53.55** +2.96 | 1.72 |
| Business | 51.58 | 1.78 | 53.36 +1.78 | 1.78 | **54.12** +2.54 | 1.78 |
| Psychology | 50.75 | 1.77 | 55.01 +4.26 | 1.76 | **56.77** +6.02 | 1.75 |
| Biology | 56.90 | 1.85 | 59.00 +2.10 | 1.84 | **61.09** +4.19 | 1.82 |

**External Transfer Results.** Our experimental results demonstrate that 🌍RG training produces meaningful improvements on established benchmarks, validating the real-world applicability of our approach. Table 3 shows that our RG-Math model achieves substantial gains on MATH [21] (+9.7%) and Big-Bench Hard [70] (+7.7%), and more marginal gains on GSM8K [11] (+0.5%). Moreover, Table 4 shows that both our RG-Math and RG-Algorithmic models significantly outperform their respective baselines over several tasks from MMLU-Pro [79]. The cross-benchmark consistency confirms 🌍RG develops transferable reasoning skills, not task-specific pattern matching.

## 5 Curriculum RLVR

Curriculum learning and related approaches [3, 23, 47, 27, 52] aim to organize the training distribution such that the learner first masters simpler instances before being exposed to progressively harder variations of a task. Ideally, such approaches result in superior task performance or faster convergence during training. In this section, we evaluate a simple form of curriculum learning during RLVR by continually increasing an 🌍RG task's complexity.

**Experimental Setup.** We train Qwen2.5-3B-Instruct [84] using GRPO [63] under two conditions: (1) curriculum learning, starting with the easiest level and progressively increasing the difficulty when performance exceeds 70% over 20 training steps, and (2) fixed difficulty, sampling uniformly from all difficulty levels. We train both models for a single epoch. For each environment, we evaluate the curriculum and non curriculum models on 50 holdout examples from each difficulty level.

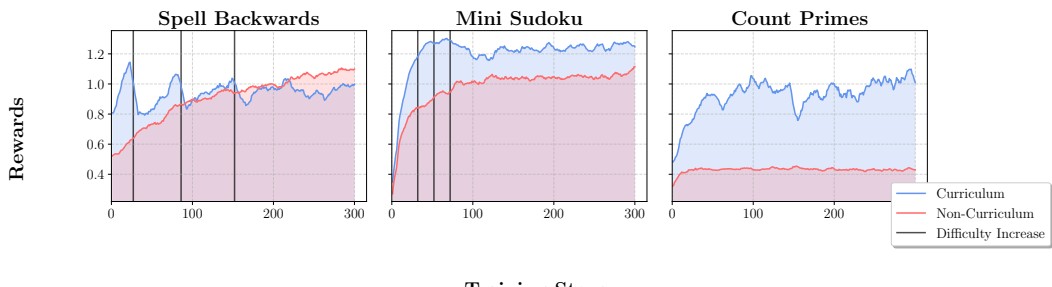

**Figure 6: Rewards for the Curriculum Learning experiments.** Reward dynamics between curriculum and non-curriculum training regimes. A vertical black line denotes a difficulty increase in curriculum training (no bars appear for Count Primes as the model never advanced past the initial level). Task difficulty increases are often proceeded by a sharp drop in reward. The curriculum model encounters increasingly difficult examples, while the non-curriculum version samples the full difficulty distribution.

**Table 5: Curriculum learning.** Baseline: Qwen2.5-3B-Instruct [84]; Non-Curriculum: Qwen2.5-3B-Instruct [84] trained on all levels concurrently; Curriculum: Qwen2.5-3B-Instruct [84] trained with an adjustable curriculum. The curriculum approach generally achieves superior performance across difficulty levels.

| Level | Baseline | Non-Curriculum | Curriculum |
|---|---|---|---|
| **Spell Backwards** (*Level = Word Length*) | | | |
| 4 | 12.00 | 30.00 | **70.67** **+40.67** |
| 6 | 3.33 | 10.67 | **30.00** **+19.33** |
| 8 | 0.00 | 1.13 | **3.37** **+2.24** |
| 10 | 0.00 | **0.01** | **0.01** **+0.00** |
| **Mini Sudoku** (*Level = # Empty Cells*) | | | |
| 4-6 | 1.13 | 54.00 | **56.00** **+2.00** |
| 6-8 | 0.00 | 25.33 | **28.00** **+2.67** |
| 8-10 | 0.00 | 6.67 | **20.00** **+13.33** |
| 10-12 | 0.00 | 1.13 | **5.33** **+4.20** |
| **Count Primes** (*Level = Number Range*) | | | |
| 100-500 | 12.00 | 4.00 | **30.67** **+26.67** |
| 100-1000 | 3.33 | 5.03 | **12.67** **+7.64** |
| 100-5000 | 1.33 | 0.00 | **3.03** **+3.03** |

**Training Dynamics.** Figure 6 compares the training dynamics of the curriculum and non-curriculum training setups. For the Spell Backward environment, increases in difficulty level are followed by a sharp drop in reward. The curriculum model's lower terminal reward reflects its exclusive exposure to maximum difficulty examples, while the non-curriculum model samples across the full difficulty distribution. In the Mini Sudoku experiment we notice that performance rises rapidly at the beginning and the model accelerates through the difficulty levels, reaching the highest level by step 72. Despite not surpassing the first difficulty level, in the Count Primes environment the curriculum approach decisively outperforms its counterpart.

**Results and Analysis.** Table 5 reveals the benefits of curriculum learning in 🌍 RG environments. The curriculum-trained models achieve superior performance to their non-curriculum equivalents in all environments and difficulty levels. There are instances in each environment where the curriculum model outperforms the non-curriculum model by significant margins e.g. $+40.67\%$ for word length of 4 on Spell Backwards, $+13.33\%$ for $8 - 10$ empty cells on Mini Sudoku and $+26.27\%$ for number range $100 - 500$ in Count Primes. Whilst Table 5 demonstrates the effectiveness of curriculum learning on 🌍 RG tasks, its value may be more limited in environments where progression paths are ambiguous or difficult to formalize.

## 6 Related Work

**Reasoning Benchmarks** There are numerous fixed dataset reasoning benchmarks. GSM8K [11], MATH, [21], OlympiadBench [20] focus on mathematical problem-solving, while BIG-Bench [66] or GPQA [55] include a diverse set of reasoning tasks. Coding benchmarks remain popular too for evaluating reasoning models [7, 26, 39, 92, 25].

However, since these benchmarks typically consist of fixed datasets, this can lead to overfitting [30, 65]. Further, benchmarks consisting of internet-scraped data are often erroneous [16, 53]. Our procedural generators differ by 1) the ability to create unlimited training examples with controllable characteristics and 2) exposing the ground-truth data-generating process. More similar to our work are procedurally generated benchmarks, e.g., in games [64, 24, 75, 33, 77, 51, 61, 87], puzzles [76, 56, 38, 91] or using generative models [57, 4, 44].

**RLVR Environments** Tülu 3 [36] builds its RL corpus by taking prompts whose answers can be objectively checked (e.g., GSM8K [11], MATH [21]) and granting a positive reward only when an automatic verifier confirms the model's answer. DeepSeek-R1 [19] kick-starts RL with a few thousand manually curated long chain-of-thought examples and then trains on automatically gradable reasoning tasks scored by rule-based accuracy and language-consistency rewards, growing the dataset to roughly 600k verified trajectories through rejection sampling.

TextArena [18] consists of 57+ text-based games suitable for both RLVR and evaluating LLMs. Logic-RL [83] procedurally generate and RLVR on logic puzzles, which generalizes cross-domain to math competition benchmarks. Similarly, Zhu et al. [91] propose a method for synthesizing open-ended logic puzzles. Zhao et al. [88] raise concerns similar to ours regarding the scarcity of high-quality, human-produced examples; however, they address this issue by proposing Absolute Zero Reasoner, a system that self-evolves its curriculum and uses a code executor to both validate proposed code reasoning tasks and verify answers. Mattern et al. [45] release the SYNTHETIC-1 reasoning dataset, including 1.4 million high-quality tasks and verifiers. OpenThoughts [71] curates reasoning datasets across various domains.

Closest to our work are parallel efforts to create libraries of reasoning environments. KORGym [60] focuses on games, providing over fifty games in textual or visual formats and including multi-turn interactions. Reasoning Core [35] is a library of procedurally generated RLVR environments across several formal domains. GEM [42] provides a standardized framework for RL environments targeted at LLMs and a diverse built-in suite of environments.

## 7 Discussion and Future Work

There are several limitations to our current approach:

- Some reasoning domains, particularly those requiring extensive domain knowledge or creativity, are difficult to capture with procedural generators. In particular, procedural generators may struggle in domains where answers are unstructured, leaving room for alternative RL data [69, 81].

- Verification functions, while comprehensive, may not capture all aspects of solution quality that humans consider important. There is still an important place for human-centric mechanisms, such as RL from human feedback [67, 50], and human-gathered RL datasets [2, 34].

- The current 🌐RG implementation focuses on single-turn, text-based reasoning and does not yet include multi-turn [80, 86] or multimodal [6, 60] reasoning tasks. Work on these is valuable to provide data for enhancing agentic and vision-language models, respectively.

- Our experiments sample data uniformly across every task, assuming independent and identically distributed data. Future work should examine continual learning settings where data arrives in non-stationary streams and investigate how regularization [31, 17, 59, 28], model merging [82, 68, 12], and replay buffers [5, 43, 54] affect the model performance under catastrophic forgetting [14, 32].

## 8 Conclusion

We have presented REASONING GYM (🌐RG), a comprehensive library of procedural dataset generators and algorithmically verifiable reasoning environments for training reasoning models with reinforcement learning. Providing over 100 tasks across diverse domains including algebra, algorithms, logic, games, and geometry, 🌐RG addresses the fundamental data scarcity bottleneck in reasoning research through unlimited, controllable task generation. Our experimental results demonstrate that RLVR training on 🌐RG tasks produces models with improved reasoning capabilities that transfer both within and across domains, as well as to established external benchmarks. Beyond training, 🌐RG serves as a rigorous evaluation framework for assessing reasoning capabilities across difficulty levels without the memorization concerns inherent to fixed benchmarks. By providing the complete library as an open-source resource, we enable the research community to systematically explore reasoning development in language models without the constraints of fixed datasets or expensive human curation.

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

# A Appendix

## A.1 Dataset categories

**Table 6:** Overview of REASONING GYM Datasets by Category

| Category (Count) | Datasets (Curriculum: ✓ yes, ✗ no) |
|---|---|
| $\sum_{ij}^x$ **Algebra** (6) *Symbolic manipulation requiring variable tracking and calculation.* | gsm_symbolic ✗, intermediate_integration ✓, polynomial_equations ✓, polynomial_multiplication ✓, simple_equations ✓, simple_integration ✓ |
| 🖳 **Algorithms** (32) *Applying procedural steps and computational thinking.* | base_conversion ✓, binary_alternation ✓, chain_sum ✓, count_bits ✓, count_primes ✓, decimal_chain_sum ✓, game_of_life ✓, game_of_life_halting ✓, gcd ✓, group_anagrams ✓, isomorphic_strings ✓, lcm ✓, letter_counting ✓, list_functions ✗, manipulate_matrix ✓, needle_haystack ✓, number_filtering ✓, number_sorting ✓, palindrome_generation ✓, palindrome_partitioning ✓, prime_factorization ✓, products ✓, ransom_note ✓, rotate_matrix ✓, spell_backward ✓, spiral_matrix ✓, string_insertion ✓, string_manipulation ✓, string_splitting ✓, string_synthesis ✓, word_sequence_reversal ✓, word_sorting ✓ |
| 🧮 **Arithmetic** (11) *Mathematical problems testing calculation and symbolic reasoning.* | basic_arithmetic ✓, bitwise_arithmetic ✓, calendar_arithmetic ✓, complex_arithmetic ✓, cryptarithm ✓, decimal_arithmetic ✓, fraction_simplification ✓, leg_counting ✓, number_format ✓, power_function ✓, time_intervals ✓ |
| 🧠 **Cognition (+ARC)** (14) *Identifying and applying transformations and rules from examples.* | acre ✗, aiw ✓, arc_1d ✓, arc_agi ✓, binary_matrix ✓, boxnet ✓, color_cube_rotation ✓, emoji_mystery ✓, family_relationships ✓, figlet_font ✓, letter_jumble ✓, rearc ✓, self_reference ✓, sentence_reordering ✓ |
| `</>` **Code** (2) *Understanding code across languages.* | bf ✓, codeio ✓ |
| 🎲 **Games** (18) *Puzzles requiring systematic thinking and constraint satisfaction.* | boxnet ✓, countdown ✓, dice ✓, futoshiki ✓, jugs ✓, knight_swap ✓, mahjong_puzzle ✓, mini_sudoku ✓, n_queens ✓, puzzle24 ✓, rubiks_cube ✓, rush_hour ✓, sokoban ✓, sudoku ✓, tower_of_hanoi ✓, tsumego ✓, word_ladder ✓, zebra_puzzles ✓ |
| △ **Geometry** (4) *Spatial reasoning challenges.* | advanced_geometry ✓, pool_matrix ✓, rectangle_count ✓, simple_geometry ✓ |
| 🕸 **Graphs** (6) *Discrete structures and traversal problems.* | course_schedule ✓, graph_color ✓, largest_island ✓, maze ✓, rotten_oranges ✓, shortest_path ✓ |
| ⋰▥ **Induction** (2) *Sequence induction tasks.* | modulo_grid ✓, number_sequence ✓ |
| ⊃ **Logic** (7) *Formal deduction environments.* | ab ✓, caesar_cipher ✓, circuit_logic ✓, knights_knaves ✓, propositional_logic ✓, quantum_lock ✓, syllogism ✓ |

## A.2 Dataset Examples

In this section of the appendix, we present a detailed overview of several representative tasks from each category included in REASONING GYM. For each task, we describe its structure, complexity parameters, and provide examples.

### A.2.1 `complex_arithmetic` (Algebra)

Find the solution of an arithmetic operation involving complex numbers.

**Default Configuration**

```
min_real = -10
max_real = 10
min_imag = -10
max_imag = 10
operations = ('+', '-', '*', '/')
operations_weights = [0.4, 0.4, 0.1, 0.1]
```

**Example Task**

```
> Question: Subtract the complex numbers: (7.0 - 7.0i) - (-5.0 + 2.0i)

> Answer: 12.0 - 9.0i

> Metadata: {
'source_dataset': 'complex_arithmetic',
'source_index': 2,
'num1': (7.0, -7.0),
'num2': (-5.0, 2.0),
'operation': '-',
'result': (12, -9),
'difficulty': {
  'min_real': -10,
  'max_real': 10,
  'min_imag': -10,
  'max_imag': 10,
  'operations_weights': [0.4, 0.4, 0.1, 0.1]
  }
}
```

### A.2.2 `spiral_matrix` (Algorithmic)

Print the elements of a matrix in spiral order.

**Default Configuration**

```
min_n = 2
max_n = 10
```

**Example Task**

```
> Question: Given a matrix, your job is to generate a list of elements
in spiral order, starting from the top-left element.

The spiral order is clockwise, starting from the top-left corner.
More precisely:
- Start from the top-left corner and move right.
- Move down towards the bottom-right corner.
- Move left towards the bottom-left corner.
- Move up towards the top-right corner.
- Repeat the steps for the inner elements of the matrix until every
```

```
  entry is visited.

  Your output should be a space-separated list of integers, e.g.
  1 2 3 4 5 6

  For the matrix below, what is the list of elements in spiral order?
  3 1 3
  2 4 9
  1 0 8

  > Answer: 3 1 3 9 8 0 1 2 4

  > Metadata: {
  'source_dataset': 'spiral_matrix',
  'source_index': 0,
  'matrix': [[3, 1, 3], [2, 4, 9], [1, 0, 8]],
  'solution': [3, 1, 3, 9, 8, 0, 1, 2, 4],
  'n': 3,
  'difficulty': {'n': (2, 10)}
  }
```

### A.2.3   `arc_1d` (ARC)

Find the solution of a 1D version of an ARC problem.

**Default Configuration**

```
  min_size = 10
  max_size = 30
  num_train = 3
```

**Example Task**

```
  > Question: Find the common rule that maps an input grid to an output
  grid, given the examples below.

  Example 1:
  Input:  0 0 0 2 9 2 3 4 4 0
  Output: 2 9 2 3 4 4 0 0 0 0

  Example 2:
  Input:  0 0 0 0 4 4 2 1 1 0
  Output: 0 4 4 2 1 1 0 0 0 0

  Example 3:
  Input:  0 0 0 7 9 4 9 1 0 0
  Output: 7 9 4 9 1 0 0 0 0 0

  Below is a test input grid. Predict the corresponding output grid by
  applying the rule you found. Describe how you derived the rule and
  your overall reasoning process in detail before you submit your answer.
  Your final answer should be just the test output grid itself.

  Input:
  0 0 0 0 0 1 5 0 0 0

  > Answer: 0 0 1 5 0 0 0 0 0 0

  > Metadata: {
  'source_dataset': 'arc_1d',
```

```
 'source_index': 0,
 'task_name': 'move_3pix_colorful_left',
 'train_examples': [
   {'input': [0, 0, 0, 2, 9, 2, 3, 4, 4, 0],
    'output': [2, 9, 2, 3, 4, 4, 0, 0, 0, 0]},
   {'input': [0, 0, 0, 0, 4, 4, 2, 1, 1, 0],
    'output': [0, 4, 4, 2, 1, 1, 0, 0, 0, 0]},
   {'input': [0, 0, 0, 7, 9, 4, 9, 1, 0, 0],
    'output': [7, 9, 4, 9, 1, 0, 0, 0, 0, 0]}],
 'test_example': {
   'input': [0, 0, 0, 0, 0, 1, 5, 0, 0, 0],
   'output': [0, 0, 1, 5, 0, 0, 0, 0, 0, 0]},
 'difficulty': {'size': (10, 30)}
 }
```

### A.2.4  `prime_factorization` (**Arithmetic**)

Factorize a given number down to its primes.

**Default Configuration**

```
min_value = 2
max_value = 1000
```

**Example Task**

```
> Question: Find the prime factorization of 656.
Write the factors separated by ×
(Example: for 12 the answer would be: 2 × 2 × 3)

> Answer: 2 × 2 × 2 × 2 × 41

> Metadata: {
'source_dataset': 'prime_factorization',
'source_index': 0,
'number': 656,
'factors': [2, 2, 2, 2, 41],
'difficulty': {'value': (2, 1000)}
 }
```

### A.2.5  `bf` (**Code**)

Find the solution of a BF (Brainf*ck) program.

**Default Configuration**

```
difficulty = 1
```

**Example Task**

```
> Question: This is a BF (Brainf*ck) computer program.
What is the output?

">[-]>[-]<>++++++++++[<++++++++++>-]<+.-.+++++.--------------.+++++++++
++++++.<"

Respond only with the exact output of the program.

> Answer: onset

> Metadata: {
```

```
 'source_dataset': 'bf',
 'source_index': 0,
 'bfit_code': '\nint main() {\n    print("onset");\n}\n',
 'bf_program': '>[-]>[-]<>++++++++++[<++++++++++>-]<+.-.
               ++++.--------------.+++++++++++++++.<',
 'difficulty': {'difficulty': 1}
 }
```

### A.2.6  `figlet_font` (Cognition)

Read the contents of text written with figlet font.

**Default Configuration**

```
static_word = None
static_font = None
min_word_len = 3
max_word_len = 7
space_letters = True
```

**Example Task**

```
> Question: What word does this say?

##    ##   ######  ##          ######   ######   ######      ##
### ###  #######  ##          ######  #######  #######    #####
#######  ##       ##           ##  ##  ##       ##   ## ##
#######  #######  ##           ##     #####    #####    ##  ##
## # ##  ##       ##           ##        ##       ##   ######
##    ##  #######  #######   ######  #######  ####### ##   ##
##    ##   ######   ######   ######  ######   ######  ##   ##

> Answer: MELISSA

> Metadata: {
 'source_dataset': 'figlet_font',
 'source_index': 1,
 'font': 'stealth_',
 'space_letters': True,
 'difficulty': {'word_len': (3, 7)}
 }
```

### A.2.7  `mini_sudoku` (Games)

Solve a mini (4x4) Sudoku puzzle.

**Default Configuration**

```
min_empty = 8
max_empty = 12
```

**Example Task**

```
> Question: In 4x4 Mini Sudoku:
- Each row must contain each number from 1-4 exactly once
- Each column must contain each number 1-4 exactly once
- Each 2x2 subgrid must contain each number 1-4 exactly once

Solve this 4x4 Mini Sudoku puzzle:
4 _ _ _
```

```
_ 3 _ _
_ 1 3 _
- - - -
```

Format your response as the puzzle above, with spaces separating each
number within a row, and newlines separating rows.

```
> Answer: 4 2 1 3
1 3 4 2
2 1 3 4
3 4 2 1

> Metadata: {
'source_dataset': 'mini_sudoku',
'source_index': 0,
'puzzle': [
  [4, 0, 0, 0],
  [0, 3, 0, 0],
  [0, 1, 3, 0],
  [0, 0, 0, 0]
],
'solution': [
  [4, 2, 1, 3],
  [1, 3, 4, 2],
  [2, 1, 3, 4],
  [3, 4, 2, 1]
],
'num_empty': 12,
'difficulty': {'empty': (8, 12)}
}
```

### A.2.8 `advanced_geometry` (Geometry)

Solve geometry-related problems.

**Default Configuration**

```
min_coord = -10
max_coord = 10
```

**Example Task**

```
> Question: For triangle with vertices A=(-1, -6), B=(4, 1), and
C=(-7, 4), determine the orthocenter (intersection of altitudes).
For all geometry problems:
1. Give coordinates in the form (x, y)
2. Round decimal answers to 3 decimal places
3. Use the degree symbol ° for angles
4. Return only the angle, coordinates, or radius as your answer.

> Answer: (0.304, -1.217)

> Metadata: {
'A': (-1, -6),
'B': (4, 1),
'C': (-7, 4),
'ortho': (7/23, -28/23),
'orthocenter_exact': ('7/23', '-28/23'),
'orthocenter_approx': (0.30434782608695654, -1.2173913043478262),
'source_dataset': 'advanced_geometry',
```

```
 'source_index': 1,
 'task_type': 'orthocenter',
 'difficulty': {'min_coord': -10, 'max_coord': 10}
 }
```

### A.2.9 `shortest_path` (Graphs)

Find the shortest path between a start and an end node.

**Default Configuration**

```
min_rows = 5
max_rows = 8
min_cols = 5
max_cols = 8
p_blocked = 0.4
```

**Example Task**

```
> Question: Your task is to find the shortest path from the start to
the destination point in a grid.

The grid is represented as a matrix with the following types of cells:
- *: your starting point
- #: your destination point
- O: an open cell
- X: a blocked cell

Therefore, you need to find the shortest path from * to #,
moving only through open cells.

You may only move in four directions: up, down, left, and right.

If there is no path from * to #, simply write "infeasible".

Your output should be a sequence of directions that leads from * to #,
e.g. right right down down up left

Now, find the length of the shortest path from * to # in the
following grid:
O X X X O
O O X X X
O O # O O
* X O O X
O X X O X

> Answer: up right right

> Metadata: {
'source_dataset': 'shortest_path',
'source_index': 0,
'matrix': [
  ['O', 'X', 'X', 'X', 'O'],
  ['O', 'O', 'X', 'X', 'X'],
  ['O', 'O', '#', 'O', 'O'],
  ['*', 'X', 'O', 'O', 'X'],
  ['O', 'X', 'X', 'O', 'X']
],
'solution': ['up', 'right', 'right'],
```

```
  'difficulty': {'rows': (5, 8), 'cols': (5, 8)}
 }
```

### A.2.10 `acre` (Induction)

Determine whether new combinations of objects will activate a detector using prior observations.

**Default Configuration**

```
 train = 1
```

**Example Task**

```
> Question: You are a researcher studying causal relationships using
Blicket experiments. In these experiments, certain objects (called
'blickets') have the hidden property of activating a detector,
causing its light to turn on.

Each example shows the results of placing different combinations of
objects on the detector. Each object is described by color, material
and shape. Your task is to determine whether a new combination of
objects will cause the detector to activate.

After observing the previous examples, respond with:
- "on" if you can determine the detector light will turn on
- "off" if you can determine the detector light will stay off
- "undetermined" if there is insufficient evidence to reach a conclusion

Do not use quotation marks in your answer.

Previous experimental results:
yellow rubber cylinder → on
red rubber sphere → off
yellow rubber cylinder, red rubber sphere → on
yellow metal sphere, red metal cylinder, brown rubber cylinder,
    purple rubber sphere, yellow rubber cube → on
yellow rubber cube, brown rubber cylinder, purple rubber sphere → off
yellow metal sphere, red metal cylinder → on

New test case:
yellow rubber cylinder

What is the detector light status?

> Answer: on

> Metadata: {
'source_dataset': 'acre',
'source_index': 0
}
```

### A.2.11 `knights_knaves` (Logic)

Determine which individuals are truth-telling, and which are liars.

**Default Configuration**

```
 n_people = 2
 depth_constraint = 2
 width_constraint = 2
```

**Example Task**

```
> Question: A very special island is inhabited only by sages and fools.
Sages always tell the truth, and fools always lie.
You meet 2 inhabitants: Zoey, and Riley.
Zoey commented, "Riley is a fool".
In Riley's words: "Zoey is a sage or Riley is a sage".
So who is a sage and who is a fool?
(Format your answer like: "Zoey is a sage/fool, and Riley is a sage/fool")

> Answer: Zoey is a fool, and Riley is a sage.

> Metadata: {
'source_dataset': 'knights_knaves',
'source_index': 0,
'statements': (
   ('lying', 1), ('or', ('telling-truth', 0), ('telling-truth', 1))
),
'solution': (False, True),
'names': ['Zoey', 'Riley'],
'knight_knave_terms': {
   'knight': 'sage',
   'knave': 'fool',
   'a_knight': 'a sage',
   'a_knave': 'a fool',
   'Knight': 'Sage',
   'Knave': 'Fool'
},
'difficulty': {
   'n_people': 2,
   'depth_constraint': 2,
   'width_constraint': 2}
}
```

### A.3 Zero-Shot Evaluation: Configs

Below are the configuration files used to procedurally generate data for the zero-shot evaluation benchmarks. Each dataset lists parameters with values for the easy setting, while the hard setting values are shown in comments.

`complex_arithmetic`

```
min_real: -10  # -100
max_real: 10  # 100
min_imag: -10  # -100
max_imag: 10  # 100
operations_weights: [0.4, 0.4, 0.1, 0.1]  # [0.25, 0.25, 0.25, 0.25]
```

`intermediate_integration`

```
problem_type_weights: [0.12, 0.12, 0.12, 0.12, 0.12, 0.12, 0.12, 0.12]
                   # [0, 0, 0, 1, 0, 0, 0, 0]
```

`polynomial_equations`

```
min_degree: 1  # 2
max_degree: 3  # 3
min_terms: 2  # 3
max_terms: 4  # 4
```

`polynomial_multiplication`

```
min_terms: 2   # 4
max_terms: 4   # 8
min_value: 1   # 10
max_value: 100   # 10000
min_degree: 0   # 1
max_degree: 3   # 4
min_polynomials: 2   # 3
max_polynomials: 3   # 6
```

simple_equations

```
min_terms: 2   # 3
max_terms: 4   # 10
min_value: 1   # 10
max_value: 100   # 10000
operators_weights: [0.4, 0.4, 0.2]   # [0.35, 0.35, 0.3]
```

simple_integration

```
min_terms: 2   # 3
max_terms: 5   # 4
```

ab

```
length: 10   # 25
```

base_conversion

```
min_base: 2   # 9
max_base: 16   # 18
min_value: 0   # 10000
max_value: 1000   # 100000
```

binary_alternation

```
min_n: 10   # 50
max_n: 30   # 500
```

binary_matrix

```
p_zero: 0.25   # 0.25
min_n: 3   # 25
max_n: 10   # 50
```

caesar_cipher

```
min_rotation: 1   # 15
max_rotation: 25   # 25
min_words: 3   # 15
max_words: 20   # 25
```

count_primes

```
min_n: 1   # 10000
max_n: 10000   # 50000
```

cryptarithm

```
min_words: 2   # 5
max_words: 3   # 10
```

game_of_life

```

```
  grid_size_x: 10   # 50
  grid_size_y: 10   # 50
  filled_cells_weights: 0.1   # 0.2
  simulation_steps: 1   # 2
```

game_of_life_halting

```
  grid_size_x: 12   # 50
  grid_size_y: 12   # 50
  difficulty: 1   # 2
  num_oscillators: 5   # 7
  max_simulation_steps: 20   # 50
```

graph_color

```
  min_num_vertices: 10   # 10
  max_num_vertices: 10   # 20
  num_colors: 3   # 4
```

group_anagrams

```
  min_anagram_groups: 2   # 10
  max_anagram_groups: 10   # 50
  min_words_per_group: 2   # 2
  max_words_per_group: 5   # 5
```

isomorphic_strings

```
  min_string_length: 2   # 50
  max_string_length: 10   # 100
```

jugs

```
  num_jugs: 3   # 4
  difficulty: 10   # 10
```

letter_counting

```
  min_words: 5   # 25
  max_words: 15   # 50
```

letter_jumble

```
  min_word_len: 1   # 5
  max_word_len: 64   # 30
  min_words: 3   # 25
  max_words: 20   # 50
  min_corruption_level: 0.1   # 0.3
  max_corruption_level: 0.9   # 0.6
```

manipulate_matrix

```
  min_rows: 2   # 25
  max_rows: 10   # 50
  min_cols: 2   # 25
  max_cols: 10   # 50
  min_transforms: 1   # 3
  max_transforms: 10   # 10
```

number_filtering

```
min_numbers: 3  # 50
max_numbers: 10  # 100
min_decimals: 0  # 2
max_decimals: 4  # 4
min_value: -100.0  # -500
max_value: 100.0  # 500
```

number_sorting

```
min_numbers: 3  # 50
max_numbers: 10  # 100
min_decimals: 0  # 2
max_decimals: 2  # 4
min_value: -100.0  # -500
max_value: 100.0  # 500
```

palindrome_generation

```
min_length: 3  # 50
max_length: 10  # 100
```

palindrome_partitioning

```
min_string_len: 5  # 5
max_string_len: 15  # 15
min_substring_palindrome_len: 1  # 1
max_substring_palindrome_len: 5  # 5
```

pool_matrix

```
min_rows: 2  # 25
max_rows: 10  # 50
min_cols: 2  # 25
max_cols: 10  # 50
min_pool_size: 1  # 5
max_pool_size: 3  # 7
```

ransom_note

```
min_note_length: 1  # 50
max_note_length: 10  # 100
min_magazine_length: 2  # 100
max_magazine_length: 30  # 500
```

rotate_matrix

```
min_n: 2  # 25
max_n: 10  # 50
min_rotations: 0  # 5
max_rotations: 10  # 15
```

rotten_oranges

```
min_n: 10  # 25
max_n: 30  # 50
```

sentence_reordering

```
min_words_in_sentence: 3  # 20
max_words_in_sentence: 20  # 50
```

spell_backward

```
min_word_len: 3   # 5
max_word_len: 10  # 20
```

spiral_matrix

```
min_n: 2   # 25
max_n: 10  # 50
```

string_insertion

```
min_string_length: 5    # 50
max_string_length: 20   # 100
```

string_manipulation

```
min_string_length: 5    # 50
max_string_length: 20   # 100
```

string_splitting

```
min_initial_machines: 0   # 50
max_initial_machines: 5   # 100
```

string_synthesis

```
min_initial_blocks: 0   # 50
max_initial_blocks: 5   # 100
```

word_ladder

```
min_word_length: 4   # 3
max_word_length: 4   # 5
```

word_sequence_reversal

```
min_words: 3   # 25
max_words: 8   # 50
```

word_sorting

```
min_words: 3    # 25
max_words: 10   # 50
min_word_length: 3    # 5
max_word_length: 12   # 10
```

arc_1d

```
min_size: 10   # 25
max_size: 30   # 50
```

arc_agi

```
rotations_weights: [0.25, 0.25, 0.25, 0.25]   # [0.15, 0.3, 0.25, 0.3]
mirrors_weights: [0.2, 0.2, 0.2, 0.2, 0.2]   # [0.2, 0.2, 0.2, 0.2, 0.2]
```

rearc

```
pso_difficulty_weights: [0.14, 0.14, 0.14, 0.14, 0.14, 0.14, 0.14]
                        # [0, 0, 0, 1, 0, 0, 0]
rng_difficulty_weights: [0.14, 0.14, 0.14, 0.14, 0.14, 0.14, 0.14]
                        # [0, 0, 0, 1, 0, 0, 0]
```

basic_arithmetic

```
  min_terms: 2   # 5
  max_terms: 6   # 10
  min_digits: 1   # 2
  max_digits: 4   # 5
```

bitwise_arithmetic
```
  difficulty: 2   # 5
```

calendar_arithmetic
```
  tasks: [
      'weekday_offset',
      'weekday_of_date',
      'weekday_of_date_from_first_date',
      'recurring_event_day',
      'count_days',
      'count_business_days',
      'is_leap_year'
  ]
  # [
  #     'weekday_of_date',
  #     'is_leap_year',
  #     'weekday_offset',
  #     'count_days',
  #     'count_business_days'
  # ]
  offset_upper_bound: 100   # 200
```

chain_sum
```
  min_terms: 2   # 5
  max_terms: 6   # 8
  min_digits: 1   # 4
  max_digits: 4   # 6
```

count_bits
```
  min_n: 1   # 1000000
  max_n: 2147483647   # 100000000
```

decimal_arithmetic
```
  min_num_decimal_places: 3   # 5
  max_num_decimal_places: 3   # 8
  precision: 12   # 10
  min_terms: 2   # 5
  max_terms: 6   # 8
```

decimal_chain_sum
```
  min_terms: 2   # 5
  max_terms: 6   # 8
  min_digits: 1   # 4
  max_digits: 4   # 8
  min_decimal_places: 1   # 4
  max_decimal_places: 4   # 6
```

dice
```
  num_dice: 4   # 6
  max_dice_size: 20   # 25
```

fraction_simplification

```

```yaml
 min_value: 1   # 100
 max_value: 1000   # 1000
 min_factor: 1   # 10
 max_factor: 100   # 100
```

gcd

```yaml
 min_numbers: 2   # 3
 max_numbers: 2   # 4
 min_value: 1   # 1000
 max_value: 1000   # 10000
```

gsm_symbolic

```yaml
 # no parameters to override
```

lcm

```yaml
 min_numbers: 2   # 3
 max_numbers: 2   # 4
 min_value: 1   # 1000
 max_value: 100   # 10000
```

leg_counting

```yaml
 min_animals: 3   # 20
 max_animals: 10   # 30
 min_instances: 1   # 64
 max_instances: 15   # 256
```

number_format

```yaml
 min_num_candidates: 2   # 25
 max_num_candidates: 5   # 100
 min_n: 1000   # 100000
 max_n: 1000000000   # 1000000
 max_delta: 10.0   # 0.001
```

power_function

```yaml
 min_exponent: 0   # 4
 max_exponent: 8   # 8
```

prime_factorization

```yaml
 min_value: 2   # 1000
 max_value: 1000   # 5000
```

products

```yaml
 min_terms: 2   # 4
 max_terms: 2   # 8
 min_digits: 1   # 4
 max_digits: 5   # 8
```

time_intervals

```yaml
 max_time_difference_seconds: 86400   # 21600
 max_date_difference_days: 100   # 30
```

bf

```yaml
 difficulty: 1   # 2
```

codeio

```
  difficulty: -1  # 7
```

color_cube_rotation

```
  min_rotations: 1  # 10
  max_rotations: 3  # 50
```

figlet_font

```
  min_word_len: 3  # 5
  max_word_len: 7  # 10
```

modulo_grid

```
  size_x: 20  # 40
  size_y: 20  # 40
  max_holes: 1  # 5
  max_divisor: 20  # 7
  max_target: 20  # 3
```

needle_haystack

```
  min_num_statements: 10  # 100
  max_num_statements: 100  # 500
```

number_sequence

```
  min_terms: 4  # 5
  max_terms: 8  # 10
  min_value: -100  # -500
  max_value: 100  # 500
  max_complexity: 3  # 3
```

rectangle_count

```
  max_rectangles: 10  # 15
```

rubiks_cube

```
  cube_size: 3  # 5
  min_scramble_steps: 3  # 25
  max_scramble_steps: 10  # 50
```

countdown

```
  min_numbers: 4  # 3
  max_numbers: 6  # 9
  min_target: 100  # 100
  max_target: 999  # 1000
  min_value: 1  # 1
  max_value: 100  # 100
```

emoji_mystery

```
  min_words_in_sentence: 3  # 10
  max_words_in_sentence: 35  # 30
```

futoshiki

```
  min_board_size: 4  # 6
  max_board_size: 9  # 7
  min_difficulty: 0  # 1
  max_difficulty: 3  # 2
```

knight_swap

```
  min_nodes: 6   # 6
  max_nodes: 9   # 8
  min_pieces: 2  # 3
  max_pieces: 2  # 4
  min_steps: 4   # 1
  max_steps: 20  # 20
```

`mahjong_puzzle`

```
  min_num_rounds: 10   # 50
  max_num_rounds: 50   # 100
```

`maze`

```
  min_grid_size: 5   # 25
  max_grid_size: 10  # 50
  min_dist: 5   # 10
  max_dist: 10  # 15
```

`mini_sudoku`

```
  min_empty: 8   # 6
  max_empty: 12  # 10
```

`n_queens`

```
  n: 8   # 8
  min_remove: 1  # 4
  max_remove: 7  # 6
```

`puzzle24`

```
  min_value: 1   # 1
  max_value: 10  # 6
```

`rush_hour`

```
  min_moves: 1   # 25
  max_moves: 50  # 50
```

`sokoban`

```
  min_w: 6   # 10
  max_w: 10  # 15
  min_h: 6   # 10
  max_h: 10  # 15
```

`sudoku`

```
  min_empty: 30  # 30
  max_empty: 50  # 50
```

`tower_of_hanoi`

```
  min_disks: 3  # 5
  max_disks: 7  # 10
  min_pegs: 3   # 3
  max_pegs: 4   # 4
```

`tsumego`

```
  min_board_size: 9   # 5
  max_board_size: 13  # 15
  max_stones: 15  # 10
```

`advanced_geometry`

```
min_coord: -10   # -100
max_coord: 10   # 100
```

simple_geometry

```
min_sides: 3   # 10
max_sides: 6   # 15
```

course_schedule

```
min_num_courses: 5   # 25
max_num_courses: 10   # 50
min_num_prerequisites: 1   # 3
max_num_prerequisites: 2   # 4
min_cycle_length: 3   # 3
max_cycle_length: 5   # 4
```

family_relationships

```
min_family_size: 4   # 5
max_family_size: 8   # 9
```

largest_island

```
min_rows: 5   # 25
max_rows: 10   # 50
min_cols: 5   # 25
max_cols: 10   # 50
min_num_islands: 0   # 5
max_num_islands: 5   # 10
min_island_size: 0   # 5
max_island_size: 10   # 20
```

quantum_lock

```
difficulty: 10   # 5
```

shortest_path

```
min_rows: 5   # 25
max_rows: 8   # 50
min_cols: 5   # 25
max_cols: 8   # 50
```

acre

```
# no parameters to override
```

list_functions

```
# no parameters to override
```

aiw

```
task_type_weights: [0.33, 0.33, 0.33]   # [0.5, 0.25, 0.25]
max_entities: 6   # 10
```

circuit_logic

```
min_terms: 3   # 10
max_terms: 5   # 20
min_inputs: 2   # 4
max_inputs: 4   # 8
```

knights_knaves

```
  n_people: 2  # 3
  depth_constraint: 2  # 3
  width_constraint: 2  # 3
```

propositional_logic

```
  min_vars: 2  # 4
  max_vars: 4  # 8
  min_statements: 2  # 4
  max_statements: 4  # 8
  min_complexity: 1  # 2
  max_complexity: 3  # 4
```

self_reference

```
  difficulty: 5  # 5
```

syllogism

```
  allow_all: True  # True
  allow_no: True  # True
  allow_some: True  # False
  allow_some_not: True  # False
```

zebra_puzzles

```
  num_people: 4  # 5
  num_characteristics: 4  # 5
```

## A.4 Zero-Shot Evaluation: Per-dataset performance on Easy settings

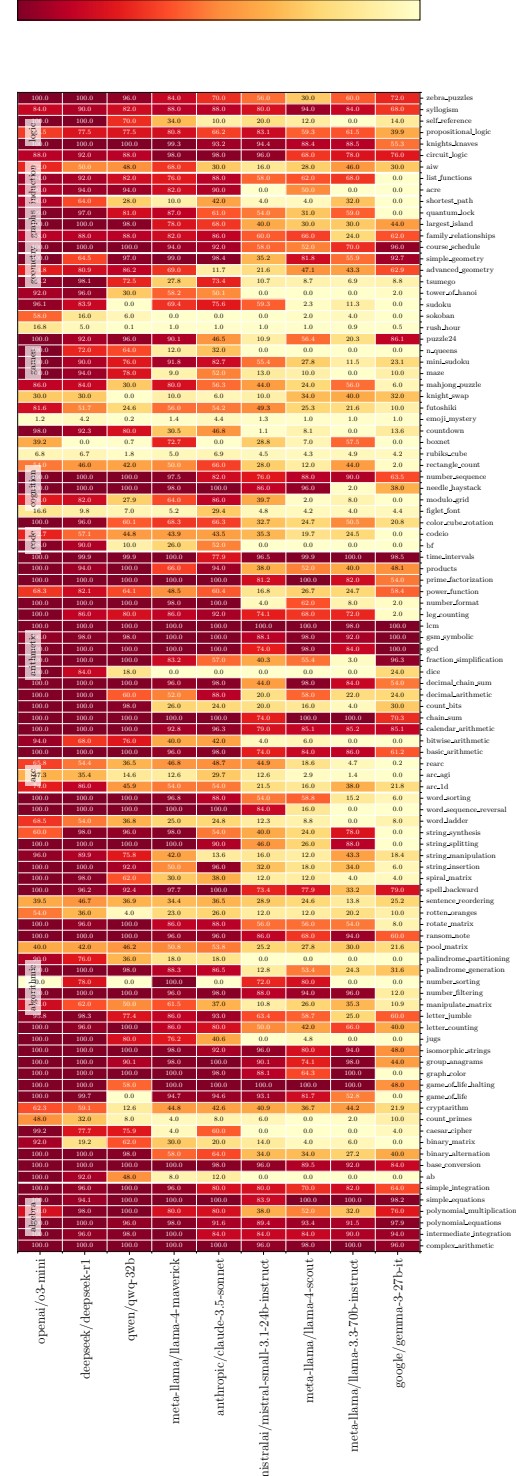

**Figure 7: Per-task reasoning accuracy on easy settings.** Top reasoning models (e.g., o3-mini, DeepSeek-R1) achieve consistently high accuracy across the majority of easy tasks, whereas leading non-reasoning baselines (e.g., Llama 4 Maverick, Claude 3.5 Sonnet) still underperform on a substantial fraction of the benchmark.

## A.5  Zero-Shot Evaluation: Per-dataset performance on Hard settings

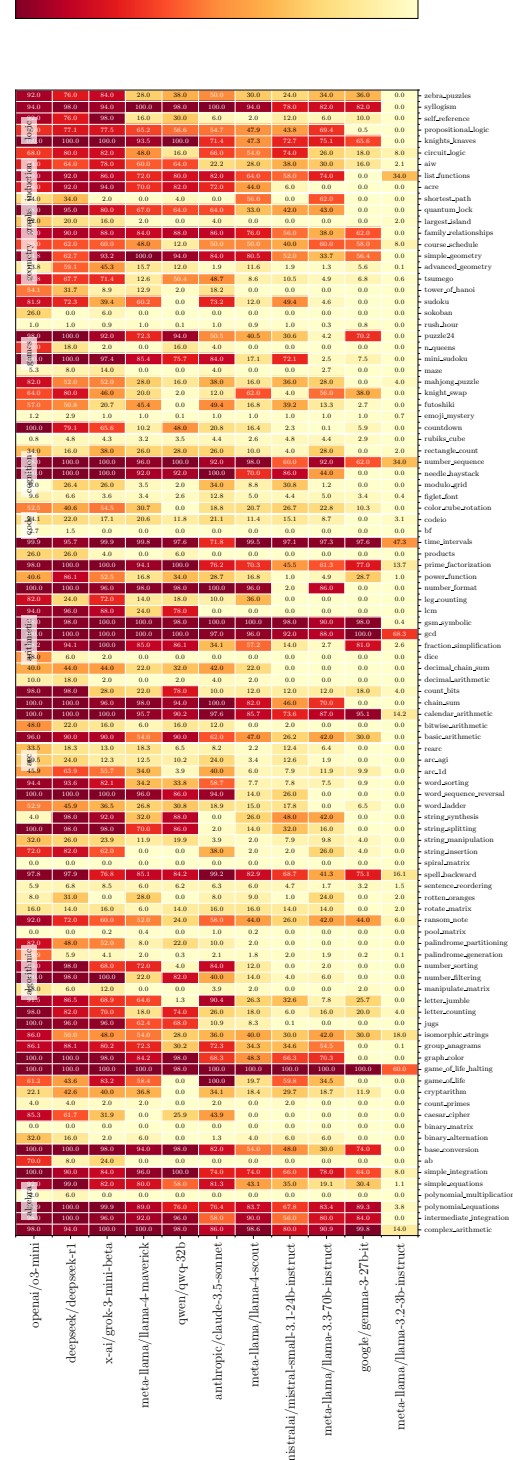

**Figure 8: Per-task reasoning accuracy on hard settings.** Performance quickly drops beyond basic skills, and even the top model (o3-mini) falters on long-horizon puzzles such as `rush_hour`, `rubiks_cube` and `rotten_oranges`, underscoring the benchmark's value for probing advanced reasoning.

## A.6 Training Approach

Here we briefly describe the approach taken to training models with RLVR. Full training code and configurations are available in our open-source repository at https://github.com/open-thought/reasoning-gym.

We use the verl open-source library for most training runs including all intra-domain and inter-domain generalization experiments, as well as curriculum learning experiments. Training runs are conducted using a 4xA6000 GPU node on Runpod.

For the Qwen2.5-3B-Instruct-RG-Math model trained as part of external generalization experiments, we use separate training code which is also included and documented in the training section of our repository, under the qwen-math subdirectory.

Below we show an example of a verl training config with custom REASONING GYM modifications. We omit two sections, reward_model and critic, which are required by verl but have no effect on the training when using GRPO due to the lack of reward and critic models.

**Example Training Config for verl**

```
reasoning_gym:
  dataset_size: 20000
  developer_prompt: DeepSeekZero
  datasets:
    ab:
      weight: 1
    base_conversion:
      weight: 1
    binary_alternation:
      weight: 1
      config:
        p_solvable: 0.9
    binary_matrix:
      weight: 1
      config:
        min_n: 2
        max_n: 6
    caesar_cipher:
      weight: 1
      config:
        max_words: 10
    cryptarithm:
      weight: 1
    isomorphic_strings:
      weight: 1
      config:
        max_string_length: 8
    jugs:
      weight: 1
      config:
        difficulty: 6
    rotate_matrix:
      weight: 1
      config:
        min_n: 2
        max_n: 6
    string_manipulation:
      weight: 1
      config:
        max_string_length: 15
        max_num_rules: 6

curriculum:
    enabled: False
    schedule:
```

```yaml
      automatic: True
      update_steps: 30 # automatic curriculum updating after 50 steps
    last_k: 20
    success_threshold: 0.70
    failure_threshold: 0.10
    curricula:
      spell_backward:
        attribute_levels:
          word_len: 0
reward:
  use_accuracy: True
  secondary_rewards:
   - name: format
     scaling_factor: 0.2
     kwargs:
        preappend_thinking_token: False
   - name: length
     scaling_factor: 0.2

data:
  tokenizer: null
  train_files: train.parquet # unused due to RG procedural dataset generators
  val_files: test.parquet # unused due to RG procedural dataset generators
  prompt_key: prompt
  max_prompt_length: 4096
  max_response_length: 2048
  train_batch_size: 32
  val_batch_size: 64
  return_raw_chat: True
  return_raw_input_ids: True
actor_rollout_ref:
  hybrid_engine: True
  model:
    path: Qwen/Qwen2.5-3B-Instruct
    external_lib: null
    override_config: { }
    enable_gradient_checkpointing: True
    use_remove_padding: True
  actor:
    strategy: fsdp  # This is for backward-compatibility
    ppo_mini_batch_size: 16
    ppo_micro_batch_size: null # will be deprecated, use
    ↪  ppo_micro_batch_size_per_gpu
    ppo_micro_batch_size_per_gpu: 8
    use_dynamic_bsz: False
    ppo_max_token_len_per_gpu: 49152 # n * ${data.max_prompt_length} +
    ↪  ${data.max_response_length}
    grad_clip: 1.0
    clip_ratio: 0.2
    entropy_coeff: 0.001
    use_kl_loss: True # True for GRPO
    kl_loss_coef: 0.001 # for grpo
    kl_loss_type: low_var_kl # for grpo
    ppo_epochs: 1
    shuffle: False
    ulysses_sequence_parallel_size: 1 # sp size
    optim:
      lr: 1e-6
      lr_warmup_steps_ratio: 0.  # the total steps will be injected during runtime
      min_lr_ratio: null   # only useful for warmup with cosine
      warmup_style: constant  # select from constant/cosine
      total_training_steps: 500  # must be override by program
    fsdp_config:
      wrap_policy:
        # transformer_layer_cls_to_wrap: None
```

```yaml
        min_num_params: 0
      param_offload: False
      optimizer_offload: False
      fsdp_size: -1
  ref:
    fsdp_config:
      param_offload: True
      wrap_policy:
        # transformer_layer_cls_to_wrap: None
        min_num_params: 0
    log_prob_micro_batch_size: null # will be deprecated, use
    ↪   log_prob_micro_batch_size_per_gpu
    log_prob_micro_batch_size_per_gpu: 160
    log_prob_use_dynamic_bsz: ${actor_rollout_ref.actor.use_dynamic_bsz}
    log_prob_max_token_len_per_gpu:
    ↪   ${actor_rollout_ref.actor.ppo_max_token_len_per_gpu}
    ulysses_sequence_parallel_size:
    ↪   ${actor_rollout_ref.actor.ulysses_sequence_parallel_size} # sp size
  rollout:
    name: vllm
    temperature: 1.0
    top_k: -1 # 0 for hf rollout, -1 for vllm rollout
    top_p: 1
    prompt_length: ${data.max_prompt_length}  # not use for opensource
    response_length: ${data.max_response_length}
    # for vllm rollout
    dtype: bfloat16 # should align with FSDP
    gpu_memory_utilization: 0.7
    ignore_eos: False
    enforce_eager: True
    free_cache_engine: True
    load_format: dummy_dtensor
    tensor_model_parallel_size: 4
    max_num_batched_tokens: 12288
    max_num_seqs: 1024
    log_prob_micro_batch_size: null # will be deprecated, use
    ↪   log_prob_micro_batch_size_per_gpu
    log_prob_micro_batch_size_per_gpu: 160
    log_prob_use_dynamic_bsz: ${actor_rollout_ref.actor.use_dynamic_bsz}
    log_prob_max_token_len_per_gpu:
    ↪   ${actor_rollout_ref.actor.ppo_max_token_len_per_gpu}
    disable_log_stats: True
    enable_chunked_prefill: True # could get higher throughput
    # for hf rollout
    do_sample: True
    use_fire_sampling: False
    max_model_len: 12288
    # number of responses (i.e. num sample times)
    n: 8 # > 1 for grpo
    val_kwargs:
      do_sample: True

algorithm:
  gamma: 1.0
  lam: 1.0
  adv_estimator: grpo
  kl_penalty: kl  # how to estimate kl divergence
  kl_ctrl:
    type: fixed
    kl_coef: 0.001

verbose: True

trainer:
  balance_batch: True
```

```yaml
total_epochs: 1
total_training_steps: 500
project_name: inter-domain-generalisation
experiment_name: inter_reasoning_algorithmic_qwen_3b_composite
logger: [ 'console', 'wandb' ]
val_generations_to_log_to_wandb: 0
nnodes: 1
n_gpus_per_node: 4
save_freq: 100
# auto: find the last ckpt to resume. If can't find, start from scratch
resume_mode: auto # or auto or resume_path if
resume_from_path: False
test_freq: 100
critic_warmup: 0
default_hdfs_dir: null
remove_previous_ckpt_in_save: False
del_local_ckpt_after_load: False
default_local_dir: checkpoints/${trainer.project_name}/${trainer.experiment_name}
```

