# OpenReview forum: "Reasoning Gym: Reasoning Environments for Reinforcement Learning with Verifiable Rewards"
_NeurIPS.cc/2025/Datasets_and_Benchmarks_Track — NeurIPS 2025 Datasets and Benchmarks Track spotlight_

### Official Review · Reviewer_AcUJ · 2025-06-16

**Rating:** 5
**Confidence:** 4

**Summary:**

This paper introduces Reasoning Gym, a library of procedurally generated, verifiable reasoning environments designed to address a key bottleneck in training large language models with reinforcement learning: the scarcity of scalable, high-quality reasoning data. Existing benchmarks are typically static and limited, making it difficult to train models that generalize. Reasoning Gym spans over 100 diverse tasks, including algebra, logic, and visual games, with automatic reward verification, adjustable difficulty, and infinite variation, enabling curriculum learning and skill transfer. The paper shows that current frontier models perform poorly on many Reasoning Gym tasks, especially as difficulty increases, and that RLVR fine-tuning with Reasoning Gym significantly improves both intra- and cross-domain generalization, as well as performance on established benchmarks like GSM8K and MATH. This work positions Reasoning Gym as a practical and extensible foundation for advancing robust reasoning in LLMs.

**Dataset Code Accessibility:**

Yes

**Dataset Code Comments:**

GItHub repository

**Ethical Considerations:**

No, there are no or only very minor ethics concerns

**Final Justification:**

The reviewers provided details during the rebuttal that partially satisfy my concerns. However, I believe this is a good paper with high potential impact and should be accepted at its current form.

**Limitations Weaknesses:**

* **W.1.** The paper is written okayish and can improve (see writing section).
* **W.2.** The RLFT domain generalization experiments are conducted on a single model.
* **W.3.** The Figures are not polished and the font size can get extremely small for som such as Figure 4(b).

**Questions**:
* **Q.1.** In line #141 its written that Figure 5 shows steep improvement except for Arithmetic category, however, it also seems to improve fast at the beginning. What am I missing here?

**Writing**:
* There are several sections which are written poorly, specifically section 2.1 with sentences such as:
   - "We also prefer tasks that can easily be varied and difficult to do via parameters."
   - "Categories reach beyond code and math-oriented verifiers typical of earlier RLVR approaches."
* Figure 1 can be better polished and presented in an easier to read manner, for example Figure 8 in the appendix is way better.

**Strengths Contributions:**

* **S.1.** The dataset covers a wide range of reasoning categories such as Algebra, Algorithmic, Arithmetic, Cognition, and Games.
* **S.2.** The dataset is evaluated on several SoTA models and is also used for RLFT with curriculum learning.
* **S.3.** Training on the dataset shows improvement on external tasks such as GSM8K and MATH.
* **S.4.** The code repository is publicly available and accessible.
* **S.5.** Section 8 is written well.

---

> ### Author Rebuttal · Authors · 2025-07-31
>
> Thank you for your review. We are glad to hear that you appreciated RG’s wide range of reasoning categories and our paper’s evaluation of SoTA models.
>
> > **W.2.** The RLFT domain generalization experiments are conducted on a single model.
>
> Agreed, we are working on including more base models for the domain generalization experiments, including Qwen-3 and other open-source models. We chose Qwen-2.5-3B-Instruct because it was the state-of-the-art open-weight model at the time of experiments and small enough to let us run dozens of RLVR experiments.
>
> > **W.3.** The Figures are not polished and the font size can get extremely small for some such as Figure 4(b).
>
> Thank you for making us aware. We acknowledge this and will revise all figures to improve clarity and ensure consistent, legible font sizes across subplots.
>
> > **Writing**: There are several sections which are written poorly, specifically section 2.1 with sentences such as:
>
> * "We also prefer tasks that can easily be varied and difficult to do via parameters."
>   * "Categories reach beyond code and math-oriented verifiers typical of earlier RLVR approaches."
> * Figure 1 can be better polished and presented in an easier to read manner, for example Figure 8 in the appendix is way better.
>
> Likewise, thank you for drawing our attention to these, we will revise those sentences and figures.
>
> Original: _"We also prefer tasks that can easily be varied and difficult to do via parameters."_
> Revised: _"We favor tasks that are both easily parameterizable yet challenging to solve."_
>
> Original: _"Categories reach beyond code and math-oriented verifiers typical of earlier RLVR approaches."_
> Revised: _"Our task categories extend beyond the code- and math-focused verifiers commonly used in prior RLVR work."_

---

> > ### Comment · Reviewer_AcUJ · 2025-08-05
> >
> > Thanks for the details. This partially satisfies my concerns and therefore I will be keeping my score. I think this paper should be accepted in it's current form.

---

### Official Review · Reviewer_s9Dt · 2025-06-29

**Rating:** 6
**Confidence:** 3

**Summary:**

This paper introduces Reasoning Gym, a library designed specifically for reinforcement learning with verifiable rewards in the domain of reasoning tasks. The authors have developed a procedural generation framework that contains more than 100 distinct reasoning environments across multiple domains, including but not limited to algebra, arithmetic, computation. The fundamental innovation of this work lies in its capacity to generate virtually unlimited training instances with adjustable complexity parameters, which represents a significant departure from traditional fixed-dataset approaches that have been predominantly employed in previous reasoning research works.

**Dataset Code Accessibility:**

Yes

**Dataset Code Comments:**

The authors provide good code accessibility through their comprehensive GitHub repository under apache license. The repo is actively maintained with recent commits and includes complete implementation with training scripts, evaluation protocols, and detailed docs.

**Ethical Comments:**

N.A.

**Ethical Considerations:**

No, there are no or only very minor ethics concerns

**Final Justification:**

My concerns have been fully addressed. This is a great contribution to the LLM RL training community.

**Limitations Weaknesses:**

(1) The external validation is quite limited, with only modest GSM8K improvements and testing on just two benchmarks (table 4). This narrow scope undermines the broader generalization claims made throughout the paper.

(2) The binary reward oversimplifies reasoning complexity, missing partial credit and multiple valid solution paths that characterize real reasoning. This limits the framework's ability to train more detailed reasoning capabilities.

(3) The cross-domain generalization results in table 3 lack theoretical analysis explaining why certain domains transfer better than others.

**Strengths Contributions:**

(1) The work provides a solid framework for RLVR, which has become quite hot in the LLM community recently. The evaluation setup has already been picked up by several key papers in the field.

(2) The procedural generation approach tackles a real bottleneck in reasoning research where fixed datasets quickly become stale and lead to overfitting. This framework lets researchers generate unlimited training data with tunable difficulty.

(3) The scope is pretty comprehensive, including  100+ task types across multiple domains from basic arithmetic to strategic games.

(4) The curriculum learning support is nicely done, allowing adaptive difficulty adjustment as models improve during training. This feature addresses a notable gap in existing benchmarks and appears essential for advancing reasoning capabilities systematically.

(5) The paper is well-organized with informative visuals e.g. table 1 and figure 2.

---

> ### Author Rebuttal · Authors · 2025-07-31
>
> Thank you for your review. We are glad to hear that you view RG as a solid framework for RLVR and appreciate its comprehensive scope and curriculum learning support. We address your concerns below:
>
>  > (1) The external validation is quite limited, with only modest GSM8K improvements and testing on just two benchmarks (table 4). This narrow scope undermines the broader generalization claims made throughout the paper.
>
> Agreed, here are some new results on MMLU-Pro and Big-Bench Hard. The RG-Math model used here is the same as that used for external generalization testing in the paper. RG-Algorithmic is trained with the same methodology but on only Algorithmic RG tasks.
>
> | Task | Qwen2.5-3B-Instruct | RG-Algorithmic | RG-Math |
> | ----- | ----- | ----- | ----- |
> | Math | 54.63 ± 1.35 | 53.89 ± 1.36 | **60.25** ± 1.33 |
> | Computer Science | 37.80 ± 2.40 | 40.73 ± 2.43 | **42.2** ± 1.47 |
> | Physics | 38.49 ± 1.35 | 39.26 ± 1.36 | **44.19** ± 1.38 |
> | Engineering | 28.28 ± 1.45 | **31.48** ± 1.49 | 31.17 ± 1.49 |
> | Economics | 50.59 ± 1.72 | 53.44 ± 1.72 | **53.55** ± 1.72 |
> | Business | 51.58 ± 1.78  | 53.36 ± 1.78 | **54.12** ± 1.78 |
> | Psychology | 50.75 ± 1.77 | 55.01 ± 1.76 | **56.77** ± 1.75 |
> | Biology | 56.90 ± 1.85 | 59.00 ± 1.84 | **61.09** ± 1.82 |
>
> | Model | Big-Bench Hard (3-shot, CoT) |
> | ----- | ----- |
> | Qwen2.5-3B-Instruct | 8.68 ± 0.30 |
> | Qwen2.5-3B-Instruct-RG-Algorithmic | 15.91 ± 0.38 |
> | Qwen2.5-3B-Instruct-RG-Math | **16.34** ± 0.40 |
>
> Crucially, the cross-domain experiments in our submission also reveal clear gains on logic- and pattern-recognition tasks that were withheld from training, evidencing generalization beyond mathematics.
>
> > (2) The binary reward oversimplifies reasoning complexity, missing partial credit and multiple valid solution paths that characterize real reasoning. This limits the framework's ability to train more detailed reasoning capabilities.
>
> Agreed, graded reward schemes are an exciting research direction, and the RG framework itself already supports partial credit for some tasks (eg, Knights and Knaves (K\&K), Rectangle Count, Sudoku).
>
> However, partial credit heuristics (e.g., edit distance, CoT overlap) often reintroduce the very subjectivity and brittleness that RLVR seeks to avoid, resulting in reward hacking. For example, in one of our partial reward K\&K experiments, the model would collapse to predict “knight” all the time because the partial reward disincentivized further exploration.
>
> Binary rewards, while simple, are not simplistic: the RG tasks with binary rewards feature rich solution spaces and tunable complexity (e.g., proof depth, constraint structure), and many require multi-step reasoning to arrive at the correct answer. As shown by consistent intra- and cross-domain transfer, even tasks with binary rewards are already sufficient to induce generalizable reasoning skills.
>
> > (3) The cross-domain generalization results in table 3 lack theoretical analysis explaining why certain domains transfer better than others.
>
> We appreciate the suggestion and agree that a theoretical account of domain transfer patterns would strengthen the work. While a full theory is out of scope, we hypothesize that domains differ in transferability based on their underlying reasoning primitives.
>
> For example, algorithmic tasks train step-wise procedural skills that naturally benefit math domains like algebra and geometry, which often decompose into similar subprocedures. Likewise, logic tasks cultivate symbolic manipulation and rule-based deduction, aiding domains like cognition and graph problems. Conversely, tasks with high visual-spatial grounding or idiosyncratic formats (e.g., ARC) offer fewer transferable abstractions. We will include these interpretations in the revision.

---

### Official Review · Reviewer_KqHz · 2025-07-02

**Rating:** 5
**Confidence:** 3

**Summary:**

This paper introduces Reasoning Gym, a comprehensive library of over 100 reasoning environments designed for reinforcement learning with verifiable rewards (RLVR). The key innovation is procedural generation of infinite training data with adjustable complexity across diverse domains including algebra, arithmetic, computation, cognition, geometry, graph theory, logic, and games. The authors demonstrate that RLVR training on these environments improves both intra-domain and inter-domain reasoning capabilities, with some transfer to external benchmarks like GSM8K and MATH.

**Additional Feedback:**

Minor issues
- Figure 3 could be larger for better readability
- The computational requirements for different task types are not well characterized

Further comments
- Provide per-category ablations, such as reward curves on different task types
- Consider releasing a task taxonomy that characterizes difficulty along dimensions like working memory, logical depth, or spatial reasoning.

**Dataset Code Accessibility:**

Yes

**Dataset Code Comments:**

It is fully accessible via an open-source GitHub repository under an Apache-2.0 license. The  procedural generators and training configurations is detailed in Section 1.

**Ethical Considerations:**

No, there are no or only very minor ethics concerns

**Final Justification:**

Given the rebuttal and the clarifications provided by the authors, I maintain my positive assessment and recommend acceptance. RLVR is a topic of growing importance in LLM. The paper makes a strong contribution by proposing a large and diverse set of reasoning environments for RLVR. Despite some limitations in generality and training scope, the core contributions are well-motivated and impactful. The rebuttal strengthened the paper.

**Limitations Weaknesses:**

-  Domains requiring extensive domain knowledge or creativity are difficult to capture procedurally, which limits the scope of reasoning types that can be addressed.
-  RL experiments are conducted primarily on 3B parameter models. Validation on larger models would strengthen the claims about RLVR effectiveness, especially given that reasoning capabilities often emerge at larger scales.
-  Only GSM8K and MATH are used for external generalization testing. Broader evaluation on diverse reasoning benchmarks e.g., logical reasoning, commonsense reasoning, would provide stronger evidence of general applicability.
-  The i.i.d. data assumption might not reflect real-world continual learning scenarios where task distributions change over time.

**Strengths Contributions:**

- The paper addresses a well-known bottleneck in reasoning research, scarcity of  diverse, and scalable data, by combining task generators with correctness-checking functions. This setup is aligned with RLVR goals and allows for continual training and evaluation.
- The level of diversity (covering 100+ tasks across 14 categories) is a strong point, particularly for measuring generalization in LLMs.
- The paper presents a series of fine-tuning experiments showing both intra-domain and inter-domain gains using GRPO, including modest improvements on GSM8K and larger gains on MATH. The curriculum learning experiments  further validate the usefulness of adaptive difficulty.
-  The code and task generators are provided,  and seems to be well-documented.

---

> ### Author Rebuttal · Authors · 2025-07-31
>
> Thank you for your review! We sincerely appreciate your acknowledgement of our paper addressing the well-known data bottleneck in RLVR research. Below, we address your comments.
>
> > Domains requiring extensive domain knowledge or creativity are difficult to capture procedurally, which limits the scope of reasoning types that can be addressed.
>
> Yes, the problem with domains requiring substantial real-world domain expertise or open-ended creativity is that they are not easily amenable to purely procedural generation. By constraining ourselves to tasks with deterministic generators, we can (i) guarantee reward correctness, (ii) avoid subjective human annotation, and (iii) study curriculum effects at scale.
>
> > RL experiments are conducted primarily on 3B parameter models. Validation on larger models would strengthen the claims about RLVR effectiveness, especially given that reasoning capabilities often emerge at larger scales.
>
> Agreed, extending experiments to larger models is valuable, and we are actively working on this for the next version of the paper. That said, we believe the current results already provide an important insight: even relatively small 3B models can acquire stronger reasoning capabilities through RLVR when trained on RG. Given the widespread assumption that reasoning only emerges at larger scales, we view this as an encouraging result; this smaller scale enables comprehensive ablations within smaller compute budgets.
>
> > Only GSM8K and MATH are used for external generalization testing. Broader evaluation on diverse reasoning benchmarks e.g., logical reasoning, commonsense reasoning, would provide stronger evidence of general applicability.
>
> Agreed, here are some new results on MMLU-Pro and Big-Bench Hard. The RG-Math model used here is the same as that used for external generalization testing in the paper. RG-Algorithmic is trained with the same methodology but on only Algorithmic RG tasks.
>
> | Task | Qwen2.5-3B-Instruct | RG-Algorithmic | RG-Math |
> | ----- | ----- | ----- | ----- |
> | Math | 54.63 ± 1.35 | 53.89 ± 1.36 | **60.25** ± 1.33 |
> | Computer Science | 37.80 ± 2.40 | 40.73 ± 2.43 | **42.2** ± 1.47 |
> | Physics | 38.49 ± 1.35 | 39.26 ± 1.36 | **44.19** ± 1.38 |
> | Engineering | 28.28 ± 1.45 | **31.48** ± 1.49 | 31.17 ± 1.49 |
> | Economics | 50.59 ± 1.72 | 53.44 ± 1.72 | **53.55** ± 1.72 |
> | Business | 51.58 ± 1.78  | 53.36 ± 1.78 | **54.12** ± 1.78 |
> | Psychology | 50.75 ± 1.77 | 55.01 ± 1.76 | **56.77** ± 1.75 |
> | Biology | 56.90 ± 1.85 | 59.00 ± 1.84 | **61.09** ± 1.82 |
>
> | Model | Big-Bench Hard (3-shot, CoT) |
> | ----- | ----- |
> | Qwen2.5-3B-Instruct | 8.68 ± 0.30 |
> | Qwen2.5-3B-Instruct-RG-Algorithmic | 15.91 ± 0.38 |
> | Qwen2.5-3B-Instruct-RG-Math | **16.34** ± 0.40 |
>
> Crucially, the cross-domain experiments in our submission also reveal clear gains on logic- and pattern-recognition tasks that were withheld from training, evidencing generalization beyond mathematics.
>
> > The i.i.d. data assumption might not reflect real-world continual learning scenarios where task distributions change over time.
>
> Yes, we have listed this as a limitation (§8 Discussion and Future work) and leave continual‑learning settings to future work because (i) frontier models already struggle with many of the i.i.d. tasks studied here, (ii) crafting realistic and interpretable distribution shifts is itself an open research problem, and (iii) building a properly curated, temporally ordered benchmark (with anti‑leakage safeguards) would require experimental complexity beyond the scope of this paper.

---

> > ### Comment · Reviewer_KqHz · 2025-08-01
> >
> > Thank you for the thorough response. The clarifications and additional results you provided strengthen the paper’s contributions and acknowledge its limitations. I maintain my positive assessment.

---

### Official Review · Reviewer_MAn8 · 2025-07-04

**Rating:** 4
**Confidence:** 3

**Summary:**

This submission introduces REASONING GYM, a comprehensive library of reinforcement learning (RL) environments designed for Reinforcement Learning with Verifiable Rewards (RLVR). Its primary contribution is a suite of over 100 procedurally generated reasoning tasks spanning diverse domains like algebra, games, logic, and cognition. A key innovation is the ability to programmatically generate a virtually infinite stream of training data with adjustable complexity, addressing the limitations of static, fixed-size reasoning datasets. The authors validate the utility of REASONING GYM through a series of experiments, including zero-shot evaluations of frontier models and analyses of intra-domain, inter-domain, and external benchmark generalization after RL-based fine-tuning.

**Dataset Code Accessibility:**

Yes

**Ethical Considerations:**

No, there are no or only very minor ethics concerns

**Final Justification:**

The authors have addressed almost all of my previous concerns, clarifying key methodological choices and strengthening the interpretation of the results. I have adjusted my score (4).

**Limitations Weaknesses:**

1. Lack of Specificity in Experimental Setup and Presentation.The paper's experimental setup lacks key details for reproducibility. For instance, it is unclear whether the main performance scores (Figure 3a) are averaged across all 100+ tasks or a specific subset, which is crucial information for interpreting the results. This, combined with minor presentation issues like the small font in Figure 3b, hinders the overall clarity of the work.

2. Insufficient Validation of Core Features.The validation for the gym's advanced features is limited. The paper claims a general curriculum learning capability (Section 3.3) but demonstrates it on only a single task (spell_backward), which is not enough to prove its universal applicability across the diverse task set. Furthermore, the underlying assumption that task parameters correlate with difficulty is asserted but not empirically validated with baseline experiments.

3. Inadequate Comparison and Differentiation from Prior Work.The paper does not sufficiently differentiate itself from closely related works like Balrog and Logic-RL, making its unique contributions less clear. It lacks a direct, feature-by-feature comparison with these existing environments. To offer the most compelling evidence of its value, a head-to-head experiment—training models on different gyms before testing on a neutral benchmark—would be highly effective but is currently absent.

**Strengths Contributions:**

1. Significant and Timely Contribution: The work addresses a critical and timely bottleneck in LLM research: the scarcity of high-quality, scalable data for improving complex reasoning. The proposed framework of using procedural generation with verifiable rewards offers a robust solution to the problem of dataset saturation and memorization that is common with static benchmarks. This is precisely the direction the field needs to move for continuous and robust model improvement.

2. Comprehensive and Diverse Task Library: A major strength is the sheer breadth and diversity of the task library. With over 100 tasks across 10 distinct categories, REASONING GYM provides a highly heterogeneous environment for developing and evaluating more general reasoning capabilities, moving beyond narrow domains. The inclusion of tasks from logic, cognition, and games, alongside more traditional math and algorithmic problems, is particularly commendable and well-illustrated in Figure 2 and Table 1.

3. Comprehensive Experimental Coverage: A key strength of this paper is the comprehensiveness of its experimental evaluation. The authors go beyond merely presenting the benchmark and conduct a wide array of experiments to validate its utility from multiple perspectives.

---

> ### Author Rebuttal · Authors · 2025-07-31
>
> Thank you for the review. We are glad you appreciate the significance, timeliness, and comprehensiveness of our paper. We respond to each of your main concerns below.
>
> > Lack of Specificity in Experimental Setup and Presentation.The paper's experimental setup lacks key details for reproducibility.
>
> This is great feedback; we will update the manuscript accordingly and ensure that more details are included.
> **The code, configuration files for reproducing all experiments, and raw results are available in our open-source repo.**
>
> > For instance, it is unclear whether the main performance scores (Figure 3a) are averaged across all 100+ tasks or a specific subset, which is crucial information for interpreting the results.
>
> Yes, the scores in Figure 3a represent averages across all 100+ RG tasks, using the hard configuration for each. These configurations are available in our repository for full reproducibility. Additionally, Appendix A.3 includes fine-grained task-level settings, and Appendix A.5 provides a heatmap of per-task performance for selected models. We appreciate the suggestion and will include more detailed plots and breakdowns in future versions of the paper.
>
> > This, combined with minor presentation issues like the small font in Figure 3b, hinders the overall clarity of the work.
>
> Thanks for drawing our attention to this; we will include higher-quality figures in the next paper iteration.
>
> > Insufficient Validation of Core Features. > The paper claims a general curriculum learning capability (Section 3.3) but demonstrates it on only a single task (spell\_backward), which is not enough to prove its universal applicability across the diverse task set.
>
> We do not want to claim universal applicability of CL, and we will soften the wording in the paper to make sure this is clear. Thank you for pointing this out.
>
> We have now run some more CL ablations across three domains in the meantime, and attached the results below. The CL strategy often outperforms the non-CL strategy, but not always.
>
> ---
>
> ### **Spell Backwards**
>
> |  | Word length |  |  |  |  |  |  |  |
> | ----- | ----- | ----- | ----- | ----- | ----- | ----- | ----- | ----- |
> | **Model** | **3** | **4** | **5** | **6** | **7** | **8** | **9** | **10** |
> | Qwen-2.5-3B-Instruct | 26.67 | 12.00 | 5.33 | 3.33 | 0.00 | 0.00 | 0.00 | 0.00 |
> | Qwen-2.5-3B-Noncurriculum | **78.00** | 30.00 | 9.33 | 10.67 | 8.67 | 1.13 | 4.00 | 0.01 |
> | Qwen-2.5-3B-Curriculum | 69.33 | **70.67** | **48.67** | **30.00** | **11.33** | **3.37** | **10.67** | **0.01** |
>
> ---
>
> ### **Mini Sudoku**
>
> |  | Number of empty cells |  |  |  |
> | ----- | ----- | ----- | ----- | ----- |
> | **Model** | **4-6** | **6-8** | **8-10** | **10-12** |
> | Qwen-2.5-3B-Instruct | 1.13 | 0.00 | 0.00 | 0.00 |
> | Qwen-2.5-3B-Noncurriculum | 54.00 | 25.33 | 6.67 | 1.13 |
> | Qwen-2.5-3B-Curriculum | **56.00** | **28.00** | **20.00** | **5.33** |
>
> ---
>
> ### **Knights and Knaves**
>
> |  | Number of People |  |  |  |
> | ----- | ----- | ----- | ----- | ----- |
> | **Model** | **2** | **3** | **4** | **5** |
> | Qwen-2.5-3B-Instruct | 14.33 | 19.66 | 4.66 | 4.66 |
> | Qwen-2.5-3B-Noncurriculum | 57.66 | 45.66 | **31.66** | **23.66** |
> | Qwen-2.5-3B-Curriculum | **59.66** | **48.66** | 30.99 | 21.33 |
>
> ---
>
> > Furthermore, the underlying assumption that task parameters correlate with difficulty is asserted but not empirically validated with baseline experiments.
>
> In Appendix A.4, we demonstrate how modulating task parameters correlates with difficulty across all domains. Please let us know if you wish to see any other experiments, and we will add them.
>
> > Inadequate Comparison and Differentiation from Prior Work. The paper does not sufficiently differentiate itself from closely related works like Balrog and Logic-RL, making its unique contributions less clear. It lacks a direct, feature-by-feature comparison with these existing environments.
>
> We appreciate the reviewer’s suggestion and will include a detailed feature-by-feature comparison in the next version of the paper. At a high level, RG is a suite of **cross-domain** tasks designed for text-based RLVR experiments. For example, the research community has already adopted RG to study whether RLVR can truly expand the reasoning capabilities of base LLMs (and the answer seems to be affirmative).
>
> In contrast, BALROG serves as an evaluation benchmark. It has been adopted to evaluate frontier VLMs, but the paper does not include any RLVR experiments.
>
> The Logic-RL paper uses the Knights\&Knaves (K\&K) task, previously introduced by Xie et al. (2024), to study whether **single-domain** RLVR can unleash cross-domain reasoning behaviour. In contrast, RG provides a suite of 104 environments, **including the same K\&K task and 103 other tasks**, to study the effects of **cross-domain** RLVR.
>
> Further, we provide a complete feature-by-feature comparison below.
>
> | Feature | ReasoningGym (RG) | BALROG | Logic‑RL |
> | ----- | ----- | ----- | ----- |
> | **\# Environments** | **104** | 6 | 1 |
> | **\# Domains** | 8 (Algorithms, Math, Games, Logic, …) | 1 (Games) | 1 (Logic) |
> | **\# Data examples** | Unlimited | Unlimited | 5,000 |
> | **RLVR experiments** | ✓ | ✗ | ✓ |
> | **Example questions of follow-up research since open-source release** | Does RL truly expand a model's reasoning capabilities? | How well do multimodal, agentic systems perform on hard games? | Does single-domain RL teach general reasoning strategies? |
>
> > To offer the most compelling evidence of its value, a head-to-head experiment—training models on different gyms before testing on a neutral benchmark—would be highly effective but is currently absent.
>
> The logic task used in Logic-RL (K\&K) is already included in RG, and we report experiments on it within our logic-related evaluations. Regarding BALROG, while we appreciate its relevance, we believe using it for training would be inappropriate, as it is currently positioned as a test-only benchmark and does not include any RL experiments. That said, our results on MATH and GSM8K already show external generalization beyond RG, and we are exploring broader benchmark comparisons in ongoing work.

---

### Decision · Program_Chairs · 2025-09-18

**Decision:**

Accept (spotlight)

**Comment:**

The primary contribution of the paper is the development of the reasoning gym, a library containing nearly 100 reasoning environments, suited for RL with verified rewards, and the ability to procedurally generate tasks and control the complexity makes it well-suited for training and testing reasoning models. The primary strength identified by all reviewers was the sheer diversity of tasks supported by the library, along with the fact that this is a very timely problem. Multiple reviewers have pointed to the limitations of the current set of experiments. They were tested on smaller models, and only a subset of tasks was used to test generalization. Additionally, a reviewer pointed out the binary nature of many of these tasks (however, that is a common feature among many reasoning problems). While I do agree with the reviewers on the limitations of the current experiment, the number of available environments is quite impressive, and I believe the availability of this library could help many researchers already working in this space. This is one of the main reasons I have recommended a spotlight, as I believe there would be significant interest in adopting this library through the community. As for the discussion, most of it was focused on the limitations of the experiments. The authors provide some additional results that seem to have resolved some of the concerns the reviewers had.